# BAM15-mediated mitochondrial uncoupling protects against obesity and improves glycemic control

Christopher L Axelrod[1,2,3] iD, William T King[1,2] iD, Gangarao Davuluri[1,4], Robert C Noland[5], Jacob Hall[1,2], Michaela Hull[3], Wagner S Dantas[1], Elizabeth RM Zunica[1,6], Stephanie J Alexopoulos[7], Kyle L Hoehn[7], Ingeborg Langohr[8], Krisztian Stadler[9], Haylee Doyle[9], Eva Schmidt[9], Stephan Nieuwoudt[3], Kelly Fitzgerald[3], Kathryn Pergola[1,2], Hisashi Fujioka[10], Jacob T Mey[1,3], Ciaran Fealy[3], Anny Mulya[3], Robbie Beyl[11], Charles L Hoppel[1,12] & John P Kirwan[1,3,6,*] iD

## Abstract

Obesity is a leading cause of preventable death worldwide. Despite this, current strategies for the treatment of obesity remain ineffective at achieving long-term weight control. This is due, in part, to difficulties in identifying tolerable and efficacious small molecules or biologics capable of regulating systemic nutrient homeostasis. Here, we demonstrate that BAM15, a mitochondrially targeted small molecule protonophore, stimulates energy expenditure and glucose and lipid metabolism to protect against diet-induced obesity. Exposure to BAM15 *in vitro* enhanced mitochondrial respiratory kinetics, improved insulin action, and stimulated nutrient uptake by sustained activation of AMPK. C57BL/6J mice treated with BAM15 were resistant to weight gain. Furthermore, BAM15-treated mice exhibited improved body composition and glycemic control independent of weight loss, effects attributable to drug targeting of lipid-rich tissues. We provide the first phenotypic characterization and demonstration of pre-clinical efficacy for BAM15 as a pharmacological approach for the treatment of obesity and related diseases.

**Keywords** AMPK; BAM15; mitochondria; obesity; type 2 diabetes
**Subject Categories** Metabolism; Pharmacology & Drug Discovery

## Introduction

Obesity is a disease that affects more than 650 million people worldwide (Ng *et al*, 2014). Upward of $150 billion is spent annually to treat acute and chronic conditions related to obesity (Finkelstein *et al*, 2009). Despite this, only four medications including orlistat, phentermine–topiramate, naltrexone–bupropion, and liraglutide are currently approved for obesity treatment (Daneschvar *et al*, 2016). Furthermore, currently available pharmacotherapy regimens rarely achieve weight loss of greater than 10% or provide patients with long-term weight control (Rucker *et al*, 2007). Limited availability and efficacy are due, in part, to the difficulty in identifying bioactive compounds with a wide therapeutic range that addresses the multifaceted causes and effects of obesity (Gadde *et al*, 2018). Interestingly, one of the earliest pharmacologic approaches to the treatment of obesity was the mitochondrial protonophore 2,4-dinitrophenol (DNP) (Tainter *et al*, 1934). DNP increases systemic energy expenditure by decreasing the proton and cation gradients of the inner mitochondrial membrane. In turn, the coupling efficiency for nutrient oxidation to ATP production is greatly reduced. Patients treated with DNP exhibit marked weight loss and improved glucose control (Simon, 1953). However, unfavorable pharmacokinetic properties, off-target effects, poor tolerability, and toxicity have limited clinical application of DNP and related compounds (Grundlingh *et al*, 2011).

Advances in small molecule screening and biochemical engineering have led to a renewed interest in mitochondrially targeted agents

---

1 Integrated Physiology and Molecular Medicine Laboratory, Pennington Biomedical Research Center, Baton Rouge, LA, USA
2 Department of Translational Services, Pennington Biomedical Research Center, Baton Rouge, LA, USA
3 Department of Inflammation and Immunity, Lerner Research Institute, Cleveland Clinic, Cleveland, OH, USA
4 Sarcopenia and Malnutrition Laboratory, Pennington Biomedical Research Center, Baton Rouge, LA, USA
5 Skeletal Muscle Metabolism Laboratory, Pennington Biomedical Research Center, Baton Rouge, LA, USA
6 Department of Nutrition, Case Western Reserve University, Cleveland, OH, USA
7 School of Biotechnology and Biomolecular Sciences, University of New South Wales, Sydney, NSW, Australia
8 Department of Pathobiological Sciences, Louisiana State University, Baton Rouge, LA, USA
9 Oxidative Stress and Disease Laboratory, Pennington Biomedical Research Center, Baton Rouge, LA, USA
10 Cryo-Electron Microscopy Core, Case Western Reserve University, Cleveland, OH, USA
11 Department of Biostatistics, Pennington Biomedical Research Center, Baton Rouge, LA, USA
12 Department of Pharmacology, Case Western Reserve University, Cleveland, OH, USA
*Corresponding author. Tel: +1 225 763 2513; E-mail: john.kirwan@pbrc.edu

for the treatment of metabolic disease (Wang *et al*, 2016). This is exemplified by reports of an extended half-life derivatization of DNP that markedly improved drug tolerability while reducing cellular toxicity (Perry *et al*, 2015). However, DNP is a weak uncoupler and depolarizes both the plasma and mitochondrial membranes (Godfraind *et al*, 1970; Juthberg & Brismar, 1997; Buckler & Vaughan-Jones, 1998), ultimately limiting clinical advancement. Recently, BAM15 ((2-fluorophenyl){6-[(2-fluorophenyl)amino](1,2,5-oxadiazolo[3,4-e]pyrazin-5-yl)}amine) was identified as a mitochondrially targeted protonophore with wide tolerability *in vitro* compared with previous generation compounds (Kenwood *et al*, 2014). However, the therapeutic potential of BAM15 in the treatment and prevention of obesity remains unknown.

Here, we provide the first evidence that BAM15 is a tolerable and efficacious protonophore that protects against diet-induced obesity. We observed that BAM15 markedly increases mitochondrial respiration and sustains activity to a greater extent than previously identified compounds. Transcriptomic profiling and protein analysis further revealed that 5' AMP-activated protein kinase (AMPK) is required for both the stability of mitochondrial function and metabolic phenotype following treatment with BAM15. We demonstrate that BAM15 is orally available, selective to lipid-rich tissues, and protects against diet-induced obesity (DIO) in C57BL/6J mice. Enhanced body weight regulation was observed concurrent with reductions in fat accrual, improvements in whole-body glucose clearance, and energy expenditure without altering body temperature. Notably, the effects of BAM15 on body composition and glycemic control were independent of reductions in body weight. Finally, we demonstrate that AMPK and acetyl-CoA carboxylase (ACC) are markedly activated in white adipose tissue (WAT) following treatment, an effect that was accompanied by morphological remodeling and reduced expression of genes regulating lipogenesis. Taken together, the data suggest that BAM15 is a small molecule mitochondrial uncoupler with therapeutic potential for the treatment of obesity and associated comorbidities.

# Results

## BAM15 improves cellular respiratory kinetics by sustained mitochondrial uncoupling

In order to determine the tolerability and function of BAM15, we first conducted drug titration and toxicology studies to optimize *in vitro* utilization. To achieve this, fully differentiated C2C12 mouse myotubes were incubated in varying concentrations of BAM15 for 16 h. We observed that BAM15 did not alter cell viability or number up to 100 μM of treatment (Fig EV1A). To contextualize these findings, we then compared Caspase 3/7 activity after overnight incubation with BAM15, DNP, and the mitochondrial uncoupler carbonyl cyanide-4-(trifluoromethoxy)phenylhydrazone (FCCP) at equimolar concentrations. DNP- and FCCP-induced Caspase 3/7 activation was present at 5 and 10 μM, respectively, which was dose-responsive (Fig 1A). BAM15 did not induce Caspase 3/7 activity up to 40 μM and was lower relative to both DNP and FCCP at all doses (Fig 1A). Next, we evaluated the respiratory kinetics of BAM15 in relation to DNP and FCCP by acutely injecting 1 μM of compound and measuring oxygen consumption

and extracellular acidification rates incrementally over a 12-h period (Fig 1B–F). Mitochondrial respiration was increased in response to acute injection with BAM15, DNP, and FCCP (Fig 1B). The cellular preference for oxidative metabolism, as evidenced by the ratio of oxygen consumption to lactate production, was also increased across all uncoupling agents (Fig 1C). This effect was most pronounced in cells exposed to BAM15 and FCCP over the 12-h period. The maximal rate of uncoupling was similar in both BAM15- and FCCP-injected cells, but notably lower after DNP (Fig 1D). The time to peak respiration was extended in cells exposed to BAM15 relative to both DNP and FCCP (Fig 1E). Furthermore, the half-life of respiratory activity was truncated in cells exposed to FCCP and DNP compared with BAM15 (Fig 1F). To determine whether the mitochondrial respiratory kinetics of BAM15 were cell type-specific, we injected 1 μM of compound into AML12 hepatocytes and 3T3-L1 adipocytes (Fig EV1B and C) and found the responses similar to those observed in C2C12 cells. To test whether the change in respiratory activity was due to uncoupling of oxidative phosphorylation (OXPHOS), we then performed studies of mitochondrial (Figs 1G–I and EV1D–J) and glycolytic (Fig EV1K and L) function in intact and permeabilized cells after 16 h of BAM15 treatment. In both intact and permeabilized cells, BAM15 increased proton leak relative to a vehicle control, which contributed to elevated intact cellular respiration (Fig 1H and I). In permeabilized cells, BAM15 increased intact cell respiration and leak in the presence of pyruvate and malate substrates (Fig 1I). In both intact and permeabilized cells (Fig 1H and I), BAM15 did not alter the maximal uncoupling rate or glycolytic function, indicative of intact electron transport (ETC.). Furthermore, increased respiration occurred independent of changes in mitochondrial content, as evidenced by both intact citrate synthase activity and mitochondrial DNA (Fig 1J and K). Taken together, these data support that *in vitro*, BAM15 is a tolerable and efficacious mitochondrial uncoupler with prolonged rates of oxygen consumption relative to previous generation compounds.

## BAM15 alters the expression of metabolic genes related to energy homeostasis

To determine whether chronic exposure to BAM15 altered metabolic signaling, we performed untargeted whole transcriptome sequencing of C2C12 cells after 16-h treatment. We identified 937 transcripts that were differentially regulated by BAM15. Of the 937 transcripts, 471 were upregulated and 466 downregulated after BAM15 treatment. We then performed hierarchical cluster analysis and visualization of the top 30 differentially expressed mRNA by BAM15 (Fig 2A). To contextualize these findings, we performed pathway analysis on differentially expressed genes. Pathways involved in fatty acid and glucose uptake, utilization, and production, as well as insulin-related signaling and function, were predominantly altered following treatment (Fig 2C). Further analysis of upstream regulators and canonical signaling revealed a molecular signature driven by activation of the insulin receptor, AMP-activated protein kinase (AMPK), protein kinase B (AKT), and glucose transporter type 4 (GLUT4) (Fig 2B). Untargeted RNA sequencing revealed a unique metabolic signature whereby recruitment of AMPK may facilitate downstream uptake of both glucose and fatty acids.

## BAM15 stimulates insulin signaling and uptake of glucose and fatty acids in an AMPK-dependent manner

Based on the RNA sequencing results, we further assessed the activation of the insulin receptor and AMPK signaling pathway in response to BAM15 (Fig 3A). We observed that under basal conditions, BAM15 increased phosphorylation of AKT substrate of 160 kDa (p-AS160$^{T642}$) and AMPK at threonine 172 (p-AMPK$^{T172}$), whereas AKT was unaltered (Fig 3B–D). Upon stimulation with 0.5 and 1 μM insulin, we found that phosphorylation of AKT at both threonine 308 (p-AKT$^{T308}$) and serine 473 (p-AKT$^{S473}$), as well as p-AS160$^{T642}$, was increased following treatment with BAM15 (Fig 3B and C). These findings were observed in parallel with enhanced plasma membrane GLUT4 translocation (Fig 3E and F). To confirm that upregulation of mRNA and protein signaling resulted in functional alterations in nutrient uptake, we then assessed nutrient utilization following 16-h treatment with BAM15 using isotopic tracer incorporation ([3-$^3$H]glucose and [1-$^{14}$C]palmitate). We found that BAM15 increased insulin-independent and insulin-dependent glucose uptake (Fig 3G) and also increased palmitate oxidation (Fig 3H). Furthermore, titration of cells with pyruvate and malate after fatty acid stimulation revealed increased metabolic flexibility in cells treated with BAM15 (Fig 3H). We then postulated that AMPK phosphorylation was required for downstream activation of AS160 and subsequent cellular nutrient uptake in response to BAM15. To test this hypothesis, we generated an AMPK-targeted shRNA construct (shAMPK) in C2C12 myoblasts (Fig 3I) and assessed mitochondrial bioenergetics, glucose uptake, and insulin signaling. Compared to empty vector (EV), shAMPK cells displayed reduced basal oxygen consumption rates (Fig 3J and K). EV cells treated with BAM15 displayed increased basal consumption similar to wild-type cells, whereas the shAMPK cells remained unchanged (Fig 3J and K). ATP-linked respiration was reduced in shAMPK compared with EV cells, unaltered by BAM15 treatment in EV, and was ablated in shAMPK cells (Fig 3J and K). The rate of proton leak was unchanged by shAMPK and increased by BAM15 to a similar extent in both EV and shAMPK cells (Fig 3J and K). Maximal uncoupled respiration was similar in EV and shAMPK cells but reduced in shAMPK cells treated with BAM15 (Fig 3J and K). Spare respiratory capacity was elevated in shAMPK cells but ablated in shAMPK cells treated with BAM15 (Fig 3J and K). These findings indicate that the activation of AMPK is downstream of induction of leak, but essential in sustaining ATP synthesis rates and electron transport activity in the presence of BAM15. EV cells treated with BAM15 displayed increased basal and insulin-stimulated glucose uptake similar to wild-type cells (Fig 3L). shAMPK cells were resistant to the insulin-independent effects of BAM15 on glucose uptake but remained sensitive to insulin stimulation (Fig 3L). We then assessed insulin signaling (Fig 3M) and observed that unlike wild type and EV, shAMPK cells displayed basal activation of p-AKT$^{T308}$, which was reduced by BAM15 treatment (Fig 3N). Notably, basal activation of p-AS160$^{T642}$ was entirely ablated in shAMPK compared with EV cells (Fig 3O). However, p-AKT$^{T308}$ and p-AS160$^{T642}$ phosphorylation was intact, albeit reduced, in shAMPK cells (Fig 3N and O). Taken together, these data suggest that BAM15 regulates nutrient uptake via insulin-independent activation of p-AS160$^{T642}$, a process that requires AMPK.

## BAM15 prevents diet-induced obesity by increasing energy expenditure in C57BL/6J mice

To determine the efficacy and feasibility of BAM15 *in vivo*, 10-week-old male diet-induced obese (DIO) C57BL/6J mice were randomized to 3 weeks of CTRL (60% HFD) or BAM15 (60% HFD + 0.1% w/w BAM15). *Ad libitum* consumption of BAM15 resulted in consumption of ~85 mg/kg/day (Fig EV2A), with peak serum concentrations of ~5 μM and a half-life of ~3 h (Fig 4A). Further assessment revealed primary distribution into adipose tissue depots and to lesser extent, liver, heart, and kidneys (Figs 4B and EV2B). By day 9, BAM15 animals displayed reduced body weight relative to control (Figs 4C and EV2C, and EV3A), an effect that persisted throughout the remainder of the treatment period. BAM15 did not affect daily (Fig EV2D and E) or cumulative food intake until day 20 (Fig 4D), at which point a mean difference in body weight of 7.5 g was observed. Neither acute intraperitoneal injection (Fig 4E) of BAM15 nor chronic oral administration (Figs 4F and EV2F) altered body temperature. Furthermore, acute oral administration of BAM15 did not alter tail heat dissipation (Fig EV2G and H). Unlike CTRL, which gained both fat and lean mass during the treatment period, BAM15 animals displayed reduced fat mass with no change in lean mass compared with baseline (Fig 4G and H). The reductions in fat mass were consistent with marked reductions in gonadal white adipose tissue (gWAT), inguinal WAT (iWAT), retroperitoneal WAT (rpWAT), and brown adipose tissue (BAT) depot weights (Fig EV3). However, there were no differences in muscle depots, such as the mixed gastrocnemius or heart (Fig EV3). In addition to adipose depots, liver weight was reduced in BAM15-treated animals (Fig EV3), along with fasting plasma glucose and insulin (Fig 4I and J). Given the reductions in fasting glucose and insulin, glycemic control was then assessed by intraperitoneal glucose tolerance testing (IPGTT). We observed that glucose clearance was improved in BAM15 animals relative to CTRL (Figs 4K and L and EV2I). To determine whether alterations in energy expenditure explained the reductions in body weight and adiposity, animals were placed in a metabolic chamber and examined over a 7-day period. Oxygen consumption and total daily energy expenditure were increased (Fig 4M and N), whereas the respiratory exchange ratio (RER) was decreased (Fig 4O) in BAM15-treated animals. Notably, the daily effect on energy expenditure was driven by changes in dark phase expenditure and consumption. In addition to energy expenditure, BAM15 preserved locomotor function, which decreased from baseline in CTRL animals (Fig EV2J and K). Cumulative water consumption was unaffected by BAM15 (Fig EV2L). To confirm that the improvements in weight regulation were attributable to increased energy expenditure, we measured fecal lipid content and observed no differences between groups (Fig 4P). Taken together, these data suggest that BAM15 protects against diet-induced obesity and improves glycemic control by increasing energy expenditure and reducing adiposity.

## BAM15 improves body composition and glycemic control independent of weight loss

Based on our observations that BAM15 improves glucose homeostasis, we sought to determine whether these changes were the result of reduced body weight. To achieve this, 10-week-old male diet-induced obese (DIO) C57BL/6J mice were randomized to 2 weeks of CTRL (60% HFD), BAM15 (60% HFD + 0.1% w/w BAM15), or a

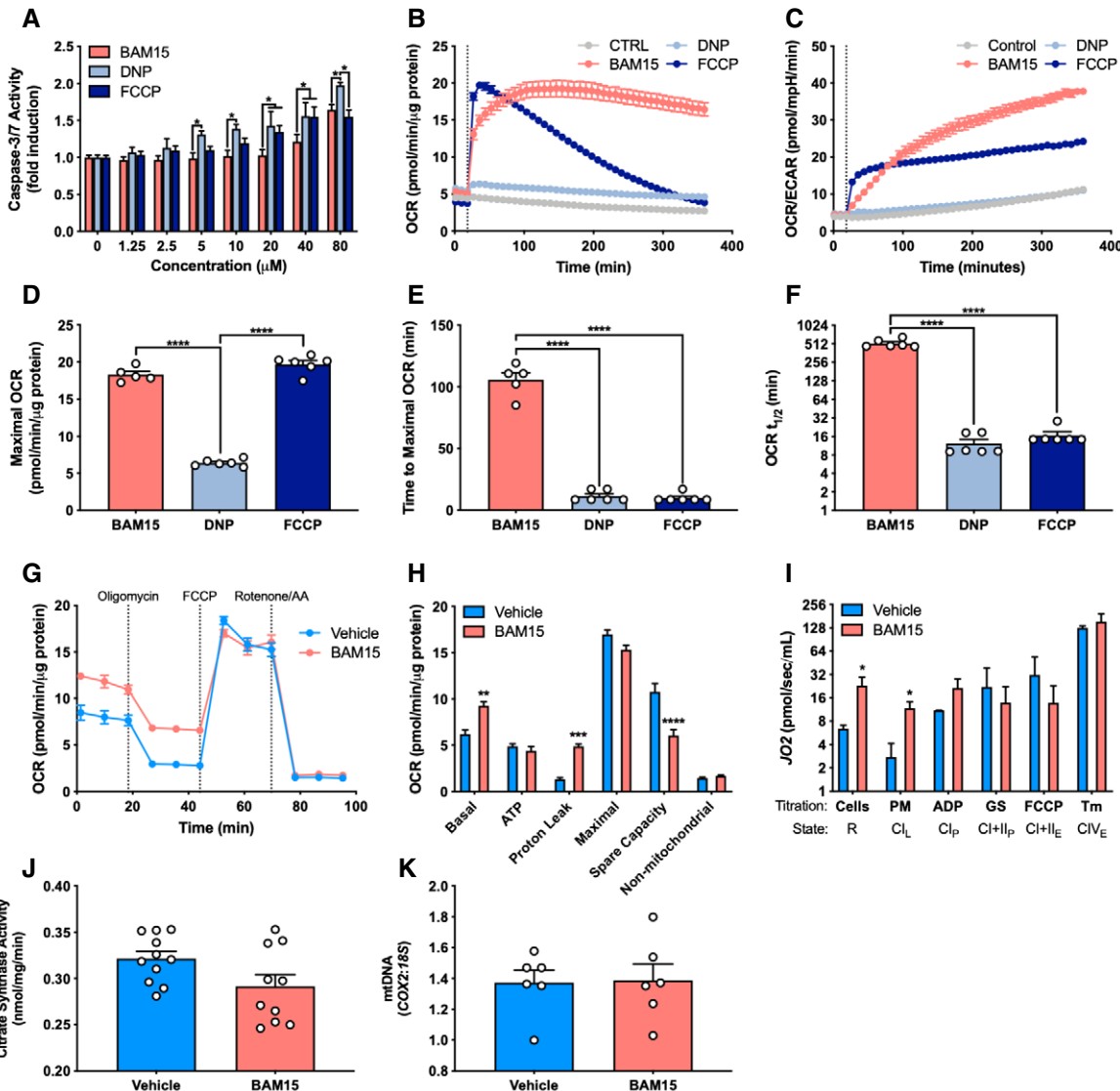

**Figure 1. BAM15 improves cellular respiratory kinetics by sustained mitochondrial uncoupling.** BAM15 improves cellular respiratory kinetics by sustained mitochondrial uncoupling.

A    Caspase 3/7 activation ± 16-h treatment with varying concentrations of BAM15, DNP, or FCCP ($N$ = 3 per condition). 5 µM: BAM15 versus DNP ($P$ = 0.030). 10 µM: BAM15 versus DNP ($P$ = 0.013). 20 µM: BAM15 versus DNP ($P$ = 0.0005), BAM15 versus FCCP ($P$ = 0.036). 40 µM: BAM15 versus DNP ($P$ = 0.018), BAM15 versus FCCP ($P$ = 0.021). 80 µM: BAM15 versus DNP ($P$ = 0.018), DNP versus FCCP ($P$ = 0.004).

B–F  (B) The rate of oxygen consumption (OCR), (C) ratio of oxygen consumption to extracellular acidification (OCR/ECAR), (D) maximal rate of OCR (BAM15 versus DNP, $P$ < 0.0001; DNP versus FCCP ($P$ < 0.0001), (E) time to maximal uncoupling rate (BAM15 versus DNP, $P$ < 0.0001; BAM15 versus FCCP ($P$ < 0.0001), and (F) mitochondrial respiratory half-life following acute injection of 1 µM BAM15, DNP, or FCCP (BAM15 versus DNP, $P$ < 0.0001; BAM15 versus FCCP) ($P$ < 0.0001) ($N$ = 6 per condition).

G, H  (G) Representative respirometry plot of intact cells ± 16-h treatment with 20 µM BAM15 sequentially injected with oligomycin, FCCP, and a cocktail of rotenone and antimycin a ($N$ = 5 per condition) and (H) basal ($P$ = 0.002) and ATP-linked respiration, proton leak ($P$ = 0.0003), maximal uncoupling, spare respiratory capacity ($P$ < 0.0001), and non-mitochondrial respiration ($N$ = 5 per condition).

I    Assessment of routine (R) oxygen flux ($P$ = 0.030), leak (L) ($P$ = 0.010), OXPHOS (P), and ETC. (E) capacity in permeabilized C2C12 cells ± 16-h treatment with BAM15 ($N$ = 4 per condition). PM: pyruvate and malate, ADP: adenosine diphosphate, GS: glutamate and succinate, Tm: tetramethyl-p-phenylenediamine.

J    Citrate synthase activity ± 16-h treatment with 20 µM BAM15 ($N$ = 10 per condition).

K    mtDNA ($COX2:18S$) ± 16-h treatment with 20 µM BAM15 ($N$ = 6 per condition).

Data information. Data are shown as the mean ± SEM. *$P$ < 0.05, **$P$ < 0.01, ***$P$ < 0.01, ****$P$ < 0.001. Panels A, H, and I were assessed by two-way repeated-measures ANOVA with Tukey's multiple comparisons. Panels D, E, and F were assessed by one-way ANOVA with Tukey's multiple comparisons. Panels J and K were assessed by unpaired Student's $t$-test.

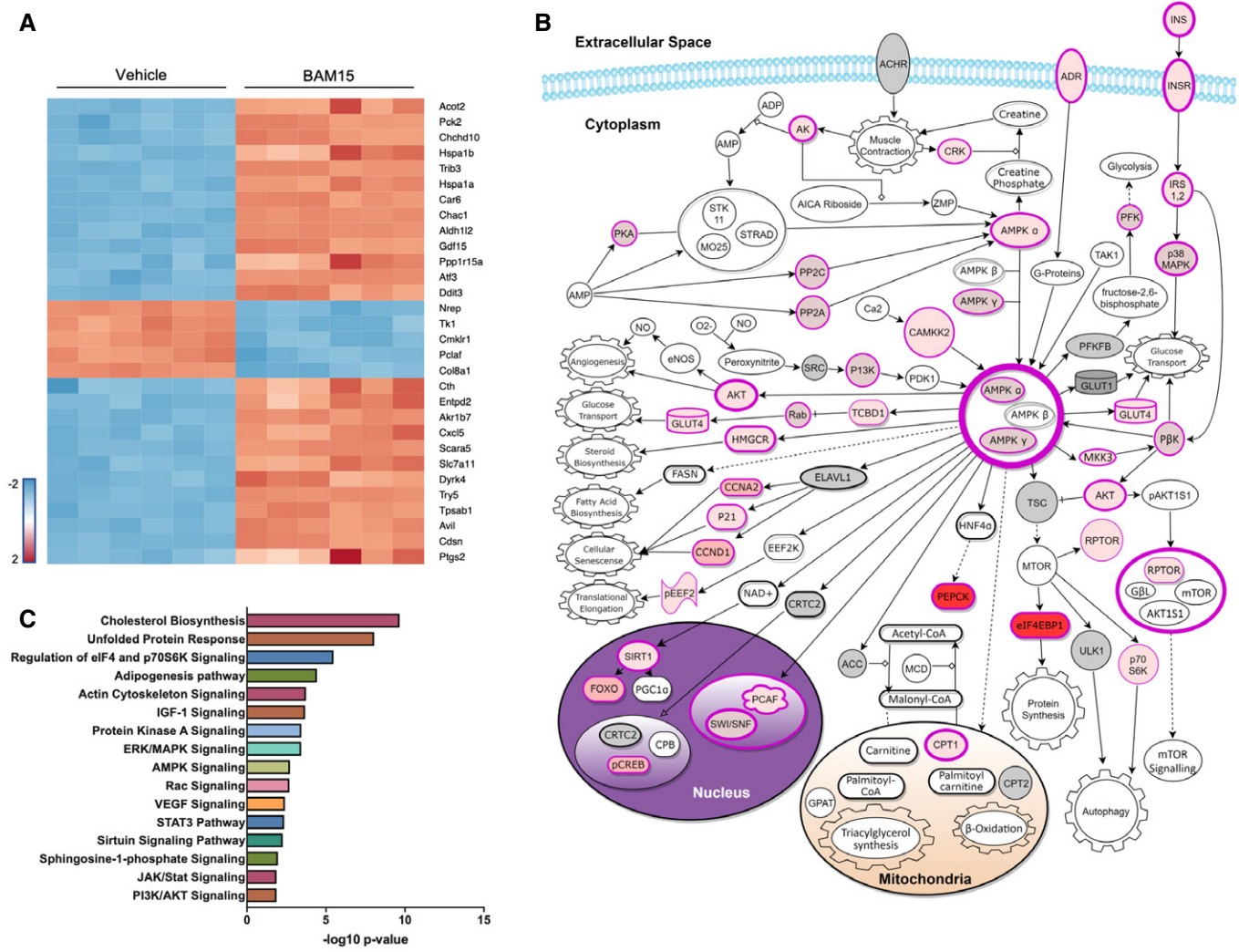

**Figure 2. BAM15 alters the expression of metabolic genes related to energy homeostasis.**

A Heat map visualization of top 30 differentially regulated mRNA transcripts. Color represents significant downregulated (blue) or upregulated (red) mRNA expressed as a log2 fold change relative to vehicle corrected for the false discovery rate.

B Schematic illustration of differentially regulated mRNA transcripts involved in the AMPK signaling pathway. Structures highlighted in shades of red were differentially regulated by BAM15. Darker shades of red correlate with increased probability of activation. Structures highlighted in gray or with black borders were unaltered by BAM15.

C Pathway enrichment analysis of differentially regulated mRNA transcripts.

calorie-restricted (CR; 60% HFD + variable restriction) HFD to reduce body weight equally to BAM15-treated animals. By day 8 of treatment, both BAM15 and CR displayed equally reduced body weight relative to CTRL animals (Fig 5A). BAM15-treated animals displayed greater reductions in fat mass relative to CR, which was increased in CTRL animals (Fig 5B). BAM15-treated animals displayed reduced lean mass relative to CTRL but increased relative to CR (Fig 5C). Daily food intake was unaltered by BAM15 but was reduced by ~15% in CR animals relative to both CTRL and BAM15 (Fig 5D). We then assessed glycemic control by IPGTT. Fasting glucose did not differ between groups (Fig 5E). BAM15 animals displayed improved glucose clearance relative to both CTRL and CR (Fig 5E and F). Fasting insulin was elevated in CTRL relative to both BAM15- and CR-treated animals (Fig 5G). Glucose-stimulated

insulin concentrations in BAM15-treated animals were reduced relative to both CTRL and CR (Fig 5G). Insulin sensitivity estimated by HOMA-IR was equally reduced in BAM15 and CR relative to CTRL-treated animals (Fig 5H). These data indicate that BAM15 improves body composition and glycemic control independent of weight loss.

## BAM15 restricts adipose tissue expansion and lipid deposition to liver and kidney in C57BL/6J mice

To determine whether any pathological features, off-target effects, or tissue-specific phenotypes of BAM15 administration were present *in vivo*, animals were necropsied after 3 weeks of treatment and tissues were fixed, paraffin-embedded, and stained with hematoxylin and eosin (H&E). Given the possibility of first-pass effect, we

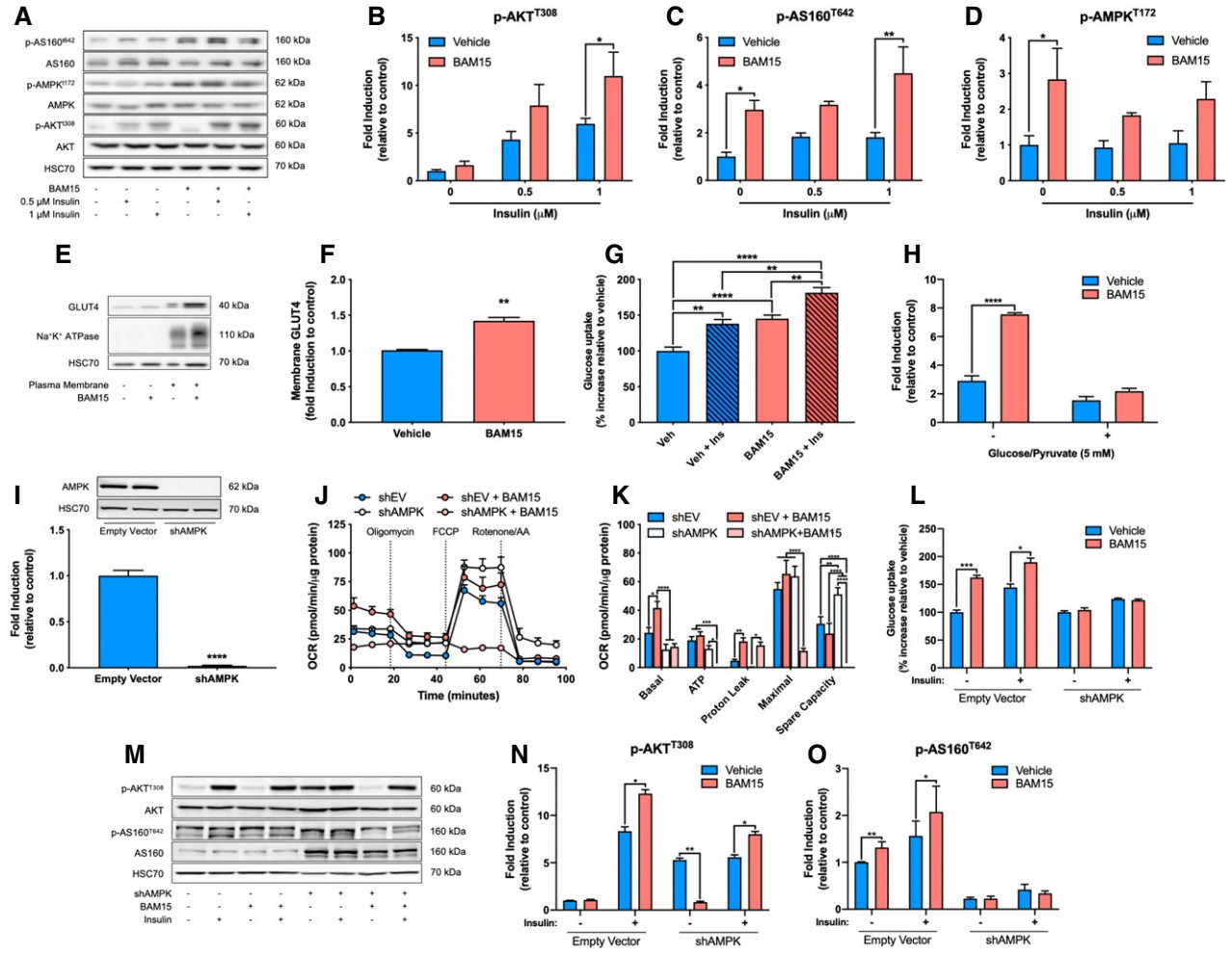

**Figure 3. BAM15 stimulates insulin signaling and oxidation of glucose and fatty acids in an AMPK-dependent manner.**

A–D  (A) Representative immunoblots and (B) quantitative analysis of AKT (1 μM insulin: $P = 0.038$), (C) AS160 (- insulin: $P = 0.036$, 1 μM insulin: $P = 0.0038$), and (D) AMPK (- insulin: $P = 0.018$) phosphorylation ± 16-h BAM15 treatment and 15-min insulin stimulation ($N = 3$ per condition) in C2C12 myotubes.

E, F  (E) Representative immunoblots and quantitative analysis of (F) plasma membrane GLUT4 translocation ($P = 0.0139$) ± 16-h BAM15 treatment ($N = 3$ per condition) in C2C12 myotubes.

G, H  (G) [3-$^3$H]glucose uptake ± 16-h BAM15 treatment and 30-min insulin stimulation ($N = 4$ per condition; vehicle versus vehicle + insulin: $P = 0.0016$, vehicle versus BAM15: $P < 0.0001$, vehicle versus BAM15 + insulin: $P < 0.0001$, vehicle + insulin versus BAM15 + insulin: $P = 0.0018$, BAM15 versus BAM15 + insulin: $P = 0.0024$) and (H) [1-$^{14}$C]palmitate oxidation ± 16-h BAM15 treatment and 5 mM glucose and 1 mM pyruvate ($N = 6$ per condition; no glucose/pyruvate: $P < 0.0001$) in C2C12 myotubes.

I  Representative immunoblot and quantitative analysis ($N = 5$ per condition; $P < 0.0001$) of AMPK demonstrating loss of function following transfection with an shAMPK construct in C2C12 myotubes.

J, K  (J) Representative respirometry plot of intact cells ± shAMPK and 16-h BAM15 treatment in C2C12 myotubes cells sequentially injected with oligomycin, FCCP, and a cocktail of rotenone and antimycin a ($N = 5$ per condition) and (K) basal and ATP-linked respiration, proton leak, maximal uncoupling, and spare respiratory capacity ($N = 5$ per condition). Basal: shEV versus shEV + BAM15 ($P = 0.0317$), shEV + BAM15 versus shAMPK ($P < 0.0001$), shEV + BAM15 versus shAMPK + BAM15 ($P < 0.0001$). ATP: shEV versus shAMPK + BAM15 ($P = 0.0026$), shEV + BAM15 versus shAMPK + BAM15 ($P = 0.0003$), shAMPK versus shAMPK + BAM15 ($P = 0.040$). Proton leak: shEV versus shEV + BAM15 ($P = 0.0052$), shAMPK versus shAMPK + BAM15 ($P = 0.0394$). Maximal: shEV versus shAMPK + BAM15 ($P < 0.0001$), shEV + BAM15 versus shAMPK + BAM15 ($P < 0.0001$), shAMPK versus shAMPK + BAM15 ($P < 0.0001$). Spare capacity: shEV versus shAMPK ($P = 0.0028$), shEV versus shAMPK + BAM15 ($P < 0.0001$), shEV + BAM15 versus shAMPK ($P < 0.0001$), shEV + BAM15 versus shAMPK + BAM15 ($P < 0.0001$), shAMPK versus shAMPK + BAM15 ($P < 0.0001$).

L  [3-$^3$H]glucose uptake ± shAMPK, 16-h BAM15 treatment, and 30-min insulin stimulation in C2C12 myotubes ($N = 3$ per condition). No insulin: shEV versus shEV + BAM15 ($P = 0.0001$). Insulin: shEV versus shEV + BAM15 ($P = 0.0142$).

M–O  (M) Representative immunoblots and quantitative analysis of (N) AKT (shEV + insulin: $P = 0.0128$, shAMPK - insulin: $P = 0.0018$, shAMPK + insulin, $P = 0.0124$), and (O) AS160 (shEV − insulin: $P = 0.0283$, shEV + insulin: $P = 0.0137$) phosphorylation ± shAMPK, 16-h BAM15 treatment, and 15-min insulin stimulation in C2C12 myotubes ($N = 3$ per condition).

Data information: Data are shown as the mean ± SEM. *$P < 0.05$, **$P < 0.01$, ***$P < 0.01$, ****$P < 0.001$. Panels B, C, D, H, K, L, N, and O were assessed by two-way repeated-measures ANOVA with Tukey's multiple comparisons. Panel G was assessed by one-way ANOVA with Tukey's multiple comparisons. Panels F and I were assessed by unpaired Student's $t$-test.

Source data are available online for this figure.

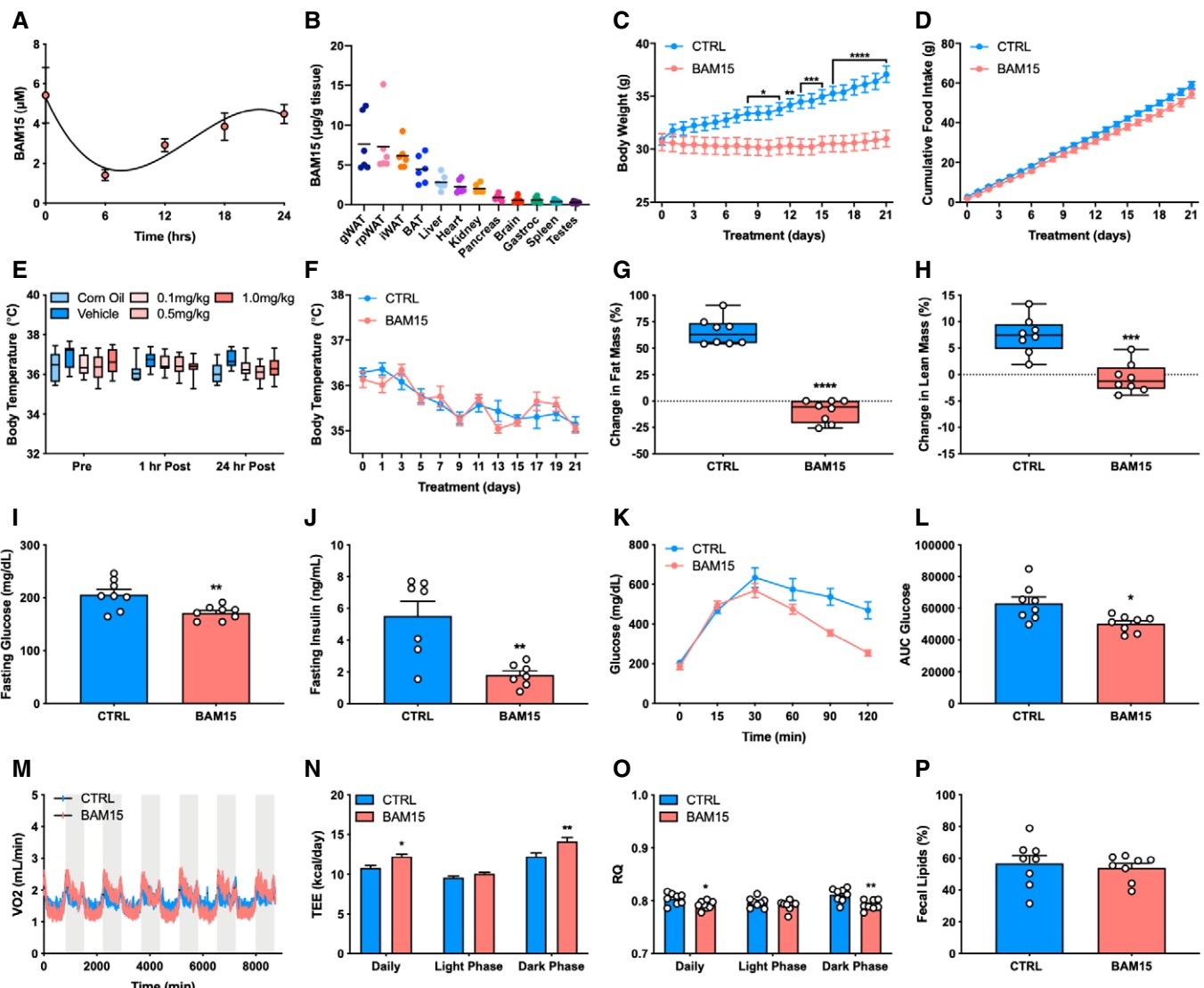

**Figure 4. BAM15 prevents diet-induced obesity by increasing energy expenditure in C57BL/6J mice.**

A, B    (A) Serum concentrations of BAM15 over a 24-h period (N = 3 animals per time point) and (B) peak tissue concentrations of BAM15 (N = 6 per tissue).
C, D    (C) Daily body weight (day 8: P = 0.0454, day 9: P = 0.0353, day 10: P = 0.0251, day 11: P = 0.0131, day 12: 0.0038, day 13: P = 0.0008, day 14: P = 0.0005, day 15: P = 0.0003, day 16–21: P < 0.0001) and (D) cumulative food intake over the 3-week treatment in CTRL- and BAM15-treated animals (N = 16 per group).
E, F    (E) Whole-body temperature before (pre), 1 h (1 h post), and 24 h (24 h post) after intraperitoneal injection of corn oil, vehicle (corn oil and saline), 0.1, 0.5, and, 1 mg BAM15 per kg of body weight (N = 11 per group), and (F) daily during chronic BAM15 treatment (N = 8 per group).
G, H    (G) Change in fat mass (%) (P < 0.0001) and (H) lean mass (%) (P < 0.0001) from baseline to 3 weeks in CTRL- and BAM15-treated animals (N = 8 per group).
I–L    (I) Fasting plasma glucose (N = 8 per group; P = 0.007) and (J) insulin (N = 7 per group; P = 0.0024) after 3 weeks of treatment in CTRL- and BAM15-treated animals. (K) Plasma glucose concentrations at 0, 15, 30, 60, 90, and 120 min following intraperitoneal injection of glucose (2 g glucose/kg of body weight; N = 8 per group) and (L) total area under the curve glucose after 3 weeks in CTRL- and BAM15-treated animals (N = 8 per group; P = 0.0106).
M–O    (M) Oxygen consumption rates (N) energy expenditure (daily: P = 0.0357, dark phase: P = 0.030), and (O) respiratory exchange ratio (daily: P = 0.0424, dark phase: P = 0.0016) measured by continuous high-resolution indirect calorimetry after 3 weeks of CTRL or BAM15 treatment (N = 8 per group).
P    Fecal energy density (% recovered in feces) after 3 weeks of CTRL or BAM15 treatment (N = 8 per group).

Data information: Data are shown as the mean ± SEM with exception to panel B with exception to panels E, H, and G which are displayed as a box (mean ± 5–95% CI) and whiskers (minimum to maximum) plot. *P < 0.05, **P < 0.01, ***P < 0.01, ****P < 0.001. Panels C, D, E, F, N, and O were assessed by two-way repeated-measures ANOVA with Tukey's multiple comparisons. Panels G, H, I, J, L, and P were assessed by an unpaired Student's t-test.

examined hepatic sections and observed large discrete vacuole formation in CTRL animals, indicative of hepatosteatosis. Vacuoles were either small or entirely absent in BAM15-treated animals (Fig 6A and C). To stage the severity of fatty liver disease, we

performed trichrome staining and observed no fibrotic development in either CTRL or BAM15 animals, suggesting specific reduction in hepatosteatosis (Fig EV4). We then assessed adipose tissue depots and found a marked reduction in the adipocyte area of iWAT,

gWAT, and BAT (Fig 6A). These changes were accompanied by increased frequency of smaller adipocytes (Fig 6G–I). Based on our observations that BAM15 markedly reduced fat accrual in both liver and adipose tissue, we assessed kidney sections by both H&E and trichrome staining. We identified elevated vacuole formation in the proximal tubules of CTRL animals, which was entirely absent in BAM15-treated animals (Fig EV4). Assessment of fibrosis by trichrome staining was negative in both CTRL- and BAM15-treated animals (Fig EV4), indicating that increased lipid deposition in CTRL animals was reflective of early-stage renal disease. Additionally, we assessed skeletal muscle sections, including both gastrocnemius and soleus tissue, and found no observable differences in nucleation, cross-sectional area, or fiber type (Fig 6A and D). Next, we evaluated pancreatic sections and found the endocrine pancreas to be generally unaffected by treatment with BAM15 (Fig EV4). Immunohistochemical staining of insulin was then performed to further examine the pancreas. We observed reduced islet hypertrophy (Fig 6B) concurrent with reductions in insulin content (Fig 6E) and β-cell mass (Fig 6F) in BAM15 relative to CTRL animals (Fig 6B) consistent with changes in circulating insulin and glycemic control. No remarkable findings were observed in the brain, reproductive organs, gastrointestinal and respiratory tract, heart, spleen, oral cavity, or rectum (Fig EV4). Taken together, these data suggest that BAM15 reduces body fat by both restricting ectopic lipid accumulation and limiting adipose tissue hypertrophy without fine or gross pathology.

### Increased skeletal muscle fatty acid oxidation is supported by AMPK-mediated suppression of lipogenesis in white adipose tissue following treatment with BAM15

Based on our observations that BAM15 reduces lipid accumulation, we then determined the tissue-specific mechanisms by which BAM15 mediates systemic energy expenditure. First, we assessed and observed a marked reduction in circulating non-esterified fatty acids (NEFAs) (Fig 7A). Plasma leptin, GDF15, and FGF21 concentrations were also reduced (Fig 7B–D), with no changes in either adiponectin or β-hydroxybutyrate (Fig EV5). We then evaluated lipid oxidation by $[1-^{14}C]$palmitate $ex\ vivo$ in mixed gastrocnemius muscle and observed increased total and $CO_2$ attributable (Fig 6E and F), but not acid-soluble (Fig EV5) oxidation in animals treated with BAM15. To determine whether the increase in oxidation was due to enhanced capacity or routine flux, we then assessed OXPHOS and ETC. function by high-resolution respirometry of permeabilized mixed gastrocnemius fibers (Fig 7G). State 3 respiration supported by glucose and fatty acid substrates was unchanged after treatment with BAM15. Furthermore, maximal ETC. capacity, determined by titration of FCCP, was also unchanged (Fig 7G). We then evaluated mitochondrial ultrastructure and content by transmission electron microscopy on mixed gastrocnemius fibers (Appendix Fig S1) and found both the intermyofibrillar and subsarcolemmal mitochondria of CTRL- and BAM15-treated animals contained similar number, distribution, and morphological characteristics (Appendix Fig S1). Based on our observations that BAM15 resulted in AMPK activation $in\ vitro$, we screened metabolically active tissues for phosphorylation activity after 3 weeks of BAM15 treatment (Fig 7H). We found that p-AMPK$^{T172}$ activity was highly upregulated in iWAT (Fig 7I and J), modestly elevated in mixed gastrocnemius muscle (Fig 7K),

and unchanged in liver and heart (Fig EV5). To confirm that p-AMPK$^{T172}$ activity was contributing to peripheral fatty acid oxidation, we then measured phosphorylation activity of acetyl-CoA carboxylase (p-ACC$^{S79}$), which was also activated in animals treated with BAM15 (Fig 7J). This led to the hypothesis that AMPK-mediated alterations in lipogenesis would result in increased free fatty acid availability for oxidation by skeletal muscle. First, we assessed lipolytic rates from both iWAT and gWAT and found no change in BAM15 relative to CTRL-treated animals (Fig EV5). We then measured gene expression of essential fatty acid metabolism regulators from iWAT and found severe downregulation of sterol regulatory element-binding transcription factor 1 ($Srebf1$; 490-fold downregulated), carbohydrate-responsive element-binding protein ($Mlxlpl$; 99-fold downregulated), fatty acid synthase ($Fasn$; 89-fold downregulated), stearoyl-CoA desaturase ($Scd1$; 898-fold downregulated), adipose triglyceride lipase ($Pnpla2$; undetected in BAM15), and peroxisome proliferator-activated receptor gamma ($Pparg$; undetected in BAM15) (Fig 7L–N). The expression of lipoprotein lipase ($Lpl$), and both the alpha and beta forms of the genes encoding for ACC ($Acaca$ and $Acacab$, respectively), remained unchanged after treatment with BAM15 (Fig 7L). Taken together, these data suggest that BAM15-mediated mitochondrial uncoupling increases whole-body energy expenditure via increased skeletal muscle fatty acid flux resulting from AMPK-mediated suppression of WAT lipogenesis.

## Discussion

Since the discovery of the protonophoric action of DNP, considerable attention has been given to the diverse application of compounds with uncoupling activity such as bupivacaine, S-13, Pcp-1, SF6847, tetrachloro-2-trifluoromethylbenzimidazole (TTFB), flufenamic acid, FCCP, and niclosamide ethanolamine (NEN) (Childress et al, 2018). However, pharmaceutical development has been limited by poor absorption, mitochondrial toxicity, variable protonophoric action, and off-target cellular effects. In this study, we provide the first evidence for the application of BAM15, a mitochondrial-targeted weak lipophilic acid with protonophore activity, as a pharmacologic strategy for the treatment of obesity, hyperglycemia, and associated comorbidities. We show that BAM15 (i) has improved mitochondrial respiratory kinetics compared with previous generation uncouplers; (ii) is orally available and shows primary distribution into the adipose and hepatic tissue of DIO male C57BL/6J mice; (iii) chronic administration protects against diet-induced obesity independent of food intake and body temperature; (iv) improves whole-body glucose disposal, body composition, and skeletal muscle oxidation of fatty acids; and (v) AMPK activation is required to sustain the metabolic action and tolerability of the compound, which $in\ vivo$ occurs primarily in adipose tissue.

The effect of mitochondrial uncoupling agents on body weight regulation is variable. DNP has been observed to reduce body weight in both humans and rodent models. However, derivatization of DNP to improve tolerability and prevent hyperthermia as well as housing in thermoneutral conditions ablates the anti-obesity action of the compound (Perry et al, 2013, 2015; Goldgof et al, 2014). The effect of uncouplers on body weight is further evidenced by mice

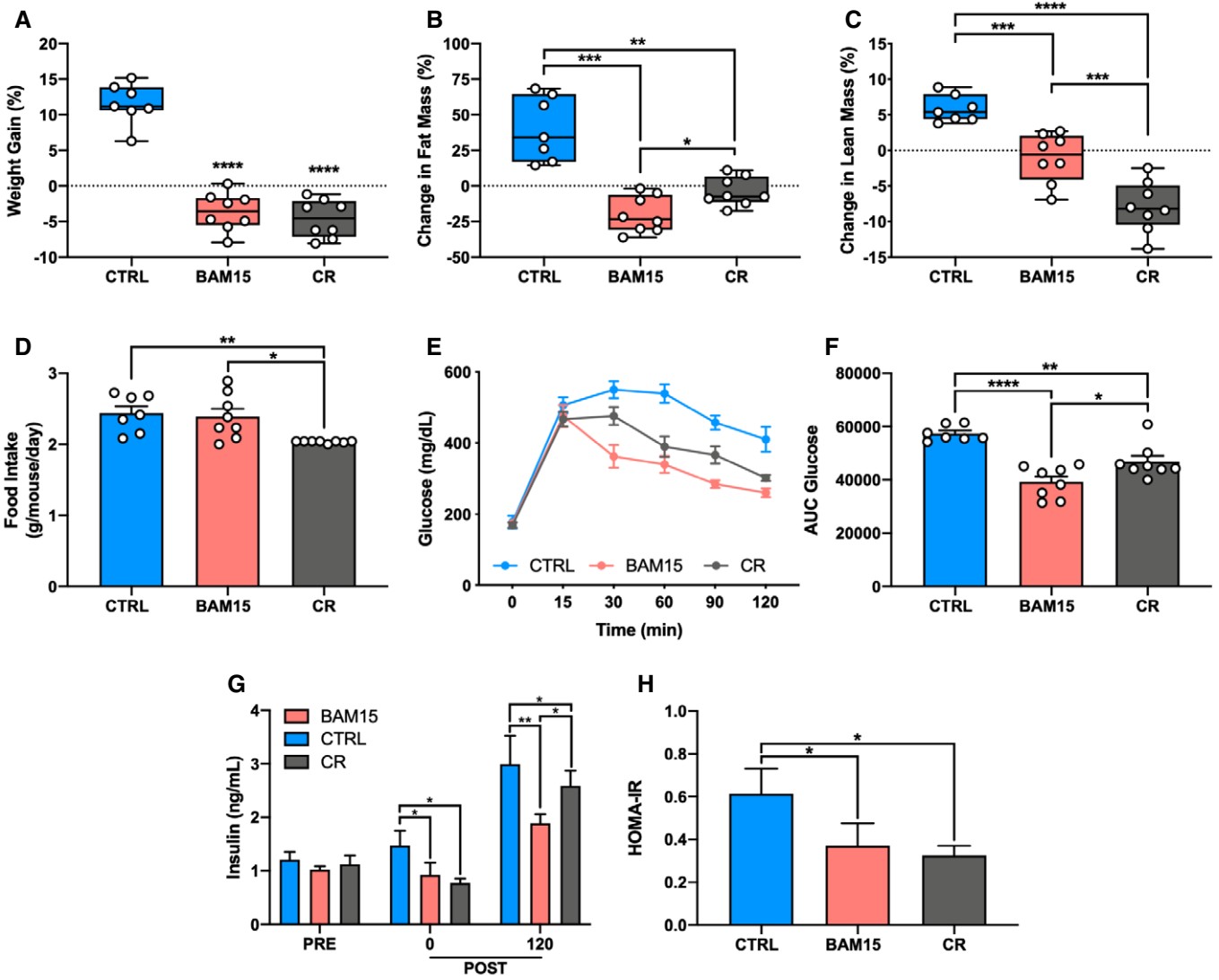

**Figure 5. BAM15 improves body composition and glycemic control independent of weight loss in C57BL/6J mice.** BAM15 improves body composition and glycemic control independent of weight loss in C57BL/6J mice.

A–D  (A) Change in body weight (CTRL versus BAM15: $P < 0.0001$, CTRL versus CR: $P < 0.0001$), (B) fat mass (CTRL versus BAM15: $P = 0.0005$, CTRL versus CR: $P = 0.0041$, BAM15 versus CR: $P = 0.043$), and (C) lean mass (CTRL versus BAM15: $P = 0.0014$, CTRL versus CR: $P < 0.0001$, BAM15 versus CR: $P = 0.0047$) from baseline to 2 weeks in CTRL-, BAM15-, and CR-treated animals and (D) daily food intake (CTRL versus CR: $P = 0.0092$, BAM15 versus CR: $P = 0.0175$) in CTRL-, BAM15-, and CR-treated animals (CTRL $N = 7$, BAM15 and CR $N = 8$).

E, F  (E) Blood glucose concentrations at 0, 15, 30, 60, 90, and 120 min following intraperitoneal injection of glucose (2 g glucose/kg of body weight; $N = 8$ per group) and (F) total area under the curve glucose (CTRL versus BAM15: $P < 0.0001$, CTRL versus CR: $P = 0.003$, BAM15 versus CR: $P = 0.0261$) after 2 weeks of treatment in CTRL-, BAM15-, and CR-treated animals (CTRL $N = 7$, BAM15 and CR $N = 8$).

G, H  (G) Plasma insulin concentrations at baseline (PRE), and 0 (CTRL versus BAM15: $P = 0.0154$, CTRL versus CR: $P = 0.0425$) and 120 min (CTRL versus BAM15: $P = 0.0005$, CTRL versus CR $P = 0.0211$, BAM15 versus CR: $P = 0.0264$) after injection of glucose after 2 weeks of treatment in CTRL-, BAM15-, and CR-treated animals (CTRL $N = 7$, BAM15 and CR $N = 8$) and (H) Homeostatic Model Assessment of Insulin Resistance (HOMA-IR) (CTRL versus BAM15: $P = 0.0259$, CTRL versus CR: $P = 0.040$) after 2 weeks of treatment in CTRL-, BAM15-, and CR-treated animals (CTRL $N = 7$, BAM15 and CR $N = 8$).

Data information: Data are shown as the mean ± SEM with exception to panels A, B, and C which are displayed as a box (mean ± 5-95% CI) and whiskers (minimum to maximum). *$P < 0.05$, **$P < 0.01$, ***$P < 0.01$, ****$P < 0.001$. Panel G was assessed by two-way repeated-measures ANOVA with Tukey's multiple comparisons. Panels A, B, C, D, F, and H were assessed by one-way ANOVA with Tukey's multiple comparisons.

treated with either Ppc-1 (Suzuki *et al*, 2015), a secondary metabolite of soil microorganisms, or NEN (Tao *et al*, 2014), an FDA-approved anthelmintic drug, both of which prevent weight gain relative to pair-fed animals. In both cases, the effect on body weight was modest and appeared secondary to metabolic improvements such as increased glycemic control and reduced liver steatosis. This is further conflated by observations in *db/db* mice where treatment with NEN increased body weight relative to control-treated animals

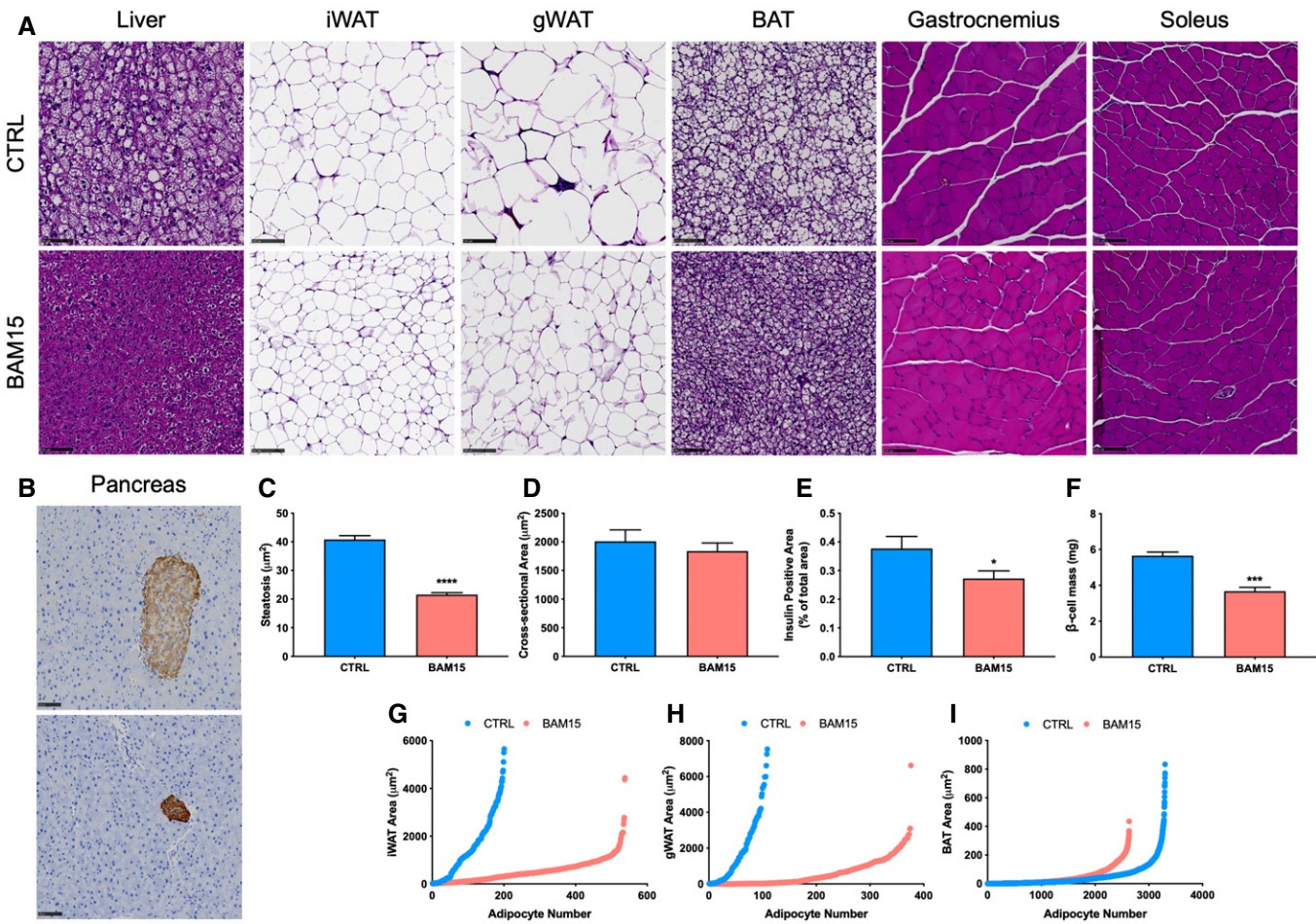

**Figure 6. BAM15 restricts adipose tissue expansion and lipid deposition to liver and kidney in C57BL/6J mice.**

A    Representative H&E-stained liver, adipose tissue, and skeletal muscle sections from CTRL- and BAM15-treated animals. Scale bars (black) = 100 µm.
B    Representative immunohistochemical analysis of pancreatic insulin (red) content counterstained with diaminobenzidine (blue). Scale bars (black) = 100 µm.
C–F  (C) Quantitative analysis of liver steatosis (µm²) (N = 6 per group; P < 0.0001), (D) gastrocnemius muscle cross-sectional area (µm²) (CTRL N = 7, BAM15 N = 8), (E) pancreatic insulin-positive area (% of total area) (N = 6 per group; P = 0.0420), (F) and β-cell mass (N = 4 per group; P = 0.0005).
G–I  Size (area; µm²) and distributional (number) analysis of (G) iWAT, (H) gWAT, and (I) BAT from CTRL- and BAM15-treated animals (N = 8 per group).

Data information: Data are shown as the mean ± SEM. *P < 0.05, ***P < 0.01, ****P < 0.001. Panels C, D, E, and F were assessed by an unpaired Student's t-test.

(Tao et al, 2014). In our experiments, administration of BAM15 prevented weight gain entirely or even modestly reduced weight and was superior to calorie restriction in both reducing adiposity and improving glycemic control. Importantly, we were unable to observe any deleterious whole body, cellular, or molecular effects on skeletal muscle tissues. Preservation of lean tissue is likely attributable to the sustained food intake observed in the BAM15-fed animals coupled with low enrichment in skeletal muscle. This is contrary to lifestyle modifications such as calorie restriction or pharmacological interventions such as phentermine, both of which achieve body weight regulation by suppression of food intake and/ or appetite, consequently reducing lean body mass (Roth et al, 2008; Kraus et al, 2019).

Parallel to protection from weight gain, we observed a ~75% reduction in hepatic lipid accumulation after treatment with BAM15 independent of AMPK activation. Inhibition of fatty acid trafficking to the liver has been consistently observed in vivo with both exogenous and endogenous activators of mitochondrial uncoupling. This

is best exemplified by DNP, where administration of the methyl ether derivatization and controlled release formulation of the compound reduce hepatosteatosis (Perry et al, 2013, 2015; Goldgof et al, 2014). Salsalate, a mild uncoupler of both hepatic and kidney mitochondria, has also been demonstrated to reduce liver fat accumulation independent of AMPK-β1 (Smith et al, 2016). The profound effect of uncoupling agents on hepatic lipid content is likely attributable to the affinity of weakly acidic lipophilic molecules to undergo first-pass hepatic clearance (Watari et al, 1988). In this view, following entry into the portal system, considerable uptake of OXPHOS uncouplers would occur, causing severe reductions in respiratory coupling efficiency, ultimately depleting hepatic lipid content. These findings are supported by our enrichment studies which demonstrated considerable uptake of BAM15 in hepatic tissue.

Passive diffusion of hydrophobic molecules into the mitochondria obscures the application of traditional pharmacokinetic profiling in that the bioactivity of the molecule is not directly

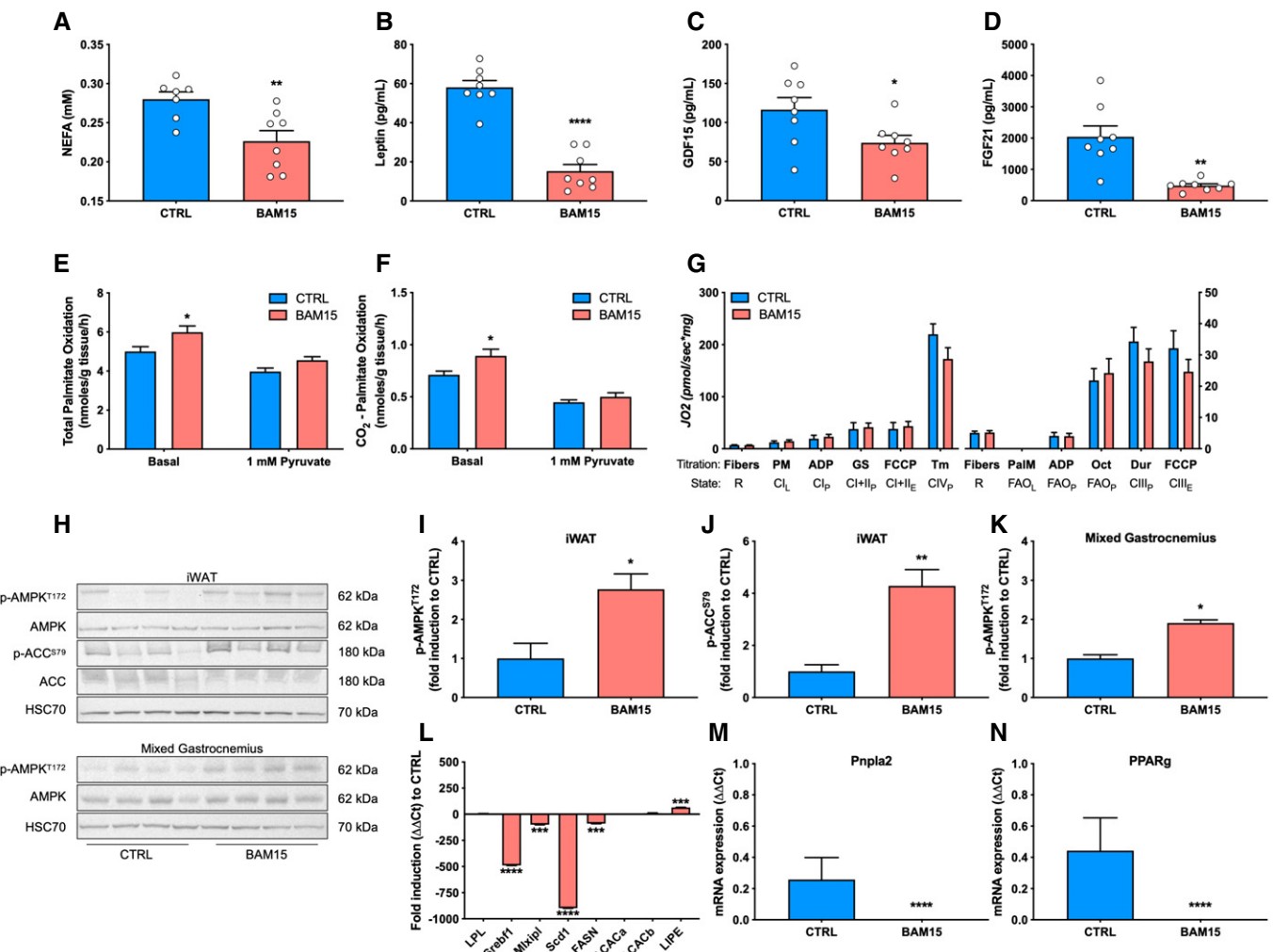

**Figure 7. Increased skeletal muscle fatty acid oxidation is supported by AMPK-mediated suppression of lipogenesis in white adipose tissue following treatment with BAM15.**

A–D  (A) Plasma concentrations of non-esterified fatty acids (NEFAs) (CTRL $N = 7$, BAM15 $N = 8$, $P = 0.0073$), (B) leptin ($N = 8$ per group, $P < 0.0001$), (C) GDF15 ($N = 7$ per group, $P = 0.035$), and (D) FGF21 ($N = 7$ per group, $P = 0.0026$) after 3 weeks of CTRL or BAM15 treatment.

E, F  (E) Total (basal: $P = 0.015$) and (F) complete (basal: $P = 0.0137$) oxidation of 200 μM [1-$^{14}$C]palmitate to $CO_2$ from mixed gastrocnemius muscle homogenates ($N = 7$ per group).

G  Routine (R) oxygen flux, leak (L), OXPHOS (P), and ETC. (E) capacity in permeabilized mixed gastrocnemius fibers ($N = 6$ per group). PM: pyruvate and malate, ADP: adenosine diphosphate, GS: glutamate and succinate, Tm: tetramethyl-p-phenylenediamine, PalM: palmitoylcarnitine and malate, Oct: octanoylcarnitine, and Dur: duroquinol.

H–K  (H) Representative immunoblots of phosphorylated and total AMPK and ACC from iWAT, and mixed gastrocnemius muscle and (I-K) densitometric quantification ($N = 4$ per group; iWAT p-AMPK$^{T172}$: $P = 0.0186$, iWAT p-ACC$^{S79}$: $P = 0.0083$, mixed gastrocnemius p-AMPK$^{T172}$: $P = 0.0286$).

L–N  mRNA expression of transcriptional regulators of lipogenesis and lipolysis in iWAT ($N = 7$ per group). Srebf1 ($P < 0.0001$), Mlxipl ($P = 0.0018$), Scd1 ($P < 0.0001$), FASN ($P = 0.0009$), LIPE ($P = 0.0015$), Pnpla2 ($P < 0.0001$), PPARg ($P < 0.0001$).

Data information: Data are shown as the mean ± SEM. *$P < 0.05$, **$P < 0.01$, ***$P < 0.01$, ****$P < 0.001$. Panels E, F, and G were assessed by two-way repeated-measures ANOVA with Tukey's multiple comparisons. Panels A, B, C, D, I, J, K, L, M, and N were assessed by an unpaired Student's $t$-test.

Source data are available online for this figure.

proportional to the systemic half-life (Zheng *et al*, 2011). In this way, quantitative subcellular analysis becomes essential in determining the organelle targeting features and function. Here, we demonstrate that despite a relatively rapid rate of disappearance from the circulation and tissues, BAM15 remains highly active in the mitochondria. The extended respiratory activity was observed in conjunction with improved cell viability and reduced cytotoxicity. It

has previously been shown that BAM15 sustains oxygen consumption over a wider concentration than FCCP in muscle and liver cells (Kenwood *et al*, 2014). Furthermore, it was demonstrated that sustained activity over a wider concentration range was due to maintaining normal plasma membrane potential, whereas FCCP caused rapid depolarization, ultimately reducing respiration at high doses (Kenwood *et al*, 2014). To our knowledge, this is the first

study to report on long-term oxygen kinetics in response to mitochondrial uncoupling. We determined *in vitro* that AMPK is activated in response to BAM15 and is required to sustain its mitochondrial function. Notably, the metabolic action *in vivo* was largely localized to WAT, where AMPK-mediated suppression of lipogenesis resulted in nutrient diversion to energy-demanding tissues, such as skeletal muscle. BAM15 has been previously shown to activate AMPK in vascular smooth muscle cells (Tai *et al*, 2018), while other mitochondrial uncouplers, such as DNP and NEN, activate AMPK in liver and cardiomyocytes, respectively (Pelletier *et al*, 2005; Tao *et al*, 2014). Recently, it was demonstrated that deletion of AMPKα in iWAT markedly reduced whole-body energy expenditure and ablated mitochondrial function (Wu *et al*, 2018). Furthermore, exposure of iWAT AMPKα-deficient mice to HFD exacerbated the development of hepatosteatosis and insulin resistance (Wu *et al*, 2018). These data further support the role of AMPK mediating the biological actions of mitochondrial protonophores by activation in response to perturbations of adenosine nucleotide ratios. Interestingly, the ability of BAM15 to increase energy expenditure in the absence of hyperthermia may be attributable to extended respiratory activity in conjunction with improved cellular tolerability relative to previous generation molecules.

Additionally, we observed a marked reduction in both circulating fatty acids and downregulation of genes contributing to *de novo* lipogenesis in WAT. These findings were further supported by primary enrichment and uptake of BAM15 into adipose tissue depots. *De novo* lipogenesis occurs primarily in hepatic and adipose tissue depots (Hollands & Cawthorne, 1981) and is highly sensitive to prevailing glucose, fat, and insulin concentrations (Kersten, 2001). In response to insulin binding, transcriptional activation of *Mlxlpl* (ChREPB) and *Srebf1* leads to fatty acid synthesis target gene activation (Iizuka *et al*, 2004; Tai *et al*, 2018). We show that both circulating insulin and iWAT expression of *Srebf1* and *Mlxlpl*, the master transcriptional regulators of lipogenesis, were reduced following treatment with BAM15. It has previously been shown that AMPK directly phosphorylates SREBP, which in turn suppresses transcriptional regulation and activation of target lipogenic genes (Li *et al*, 2011). Supporting this, *Scd1* and *Fasn*, downstream targets of ChREBP and SREBP, were also reduced after BAM15 treatment. The protein encoded by *Scd1* is responsible for converting long-chain saturated fatty acids to monounsaturated fatty acids (Ntambi *et al*, 2002), and elevated gene expression has been observed in visceral adipose tissue in humans with both obesity and T2D (Eissing *et al*, 2013). Previous investigations employing murine models demonstrated protection from dietary and genetic obesity with loss of *Scd1* (Cohen *et al*, 2002; Ntambi *et al*, 2002; Miyazaki *et al*, 2004; Jiang *et al*, 2005; MacDonald *et al*, 2008). Furthermore, overexpression of *Scd1* results in increased lipid accumulation while reducing oxidation of fatty acids (Matsui *et al*, 2012), whereas pharmacologic inhibition leads to phosphorylation of both AMPK and ACC (Kim *et al*, 2011). FASN, which converts malonyl-CoA into fatty acids, has been strongly associated with visceral fat accumulation in mice and humans (Berndt *et al*, 2007). Furthermore, treatment with the FASN inhibitor C75 markedly reduced body weight in *ob/ob* mice (Loftus *et al*, 2000). The anti-lipogenic effects of BAM15 are further supported by ablation of PPARγ gene expression after treatment. PPARγ has been widely considered one of the master regulators of adipose tissue expansion. However, loss-of-

function models and pharmacologic intervention with thiazolidinediones support a beneficial role for PPARγ agonists. Herein, suppression of PPARγ was observed in conjunction with improved insulin sensitivity and improved body composition, suggesting that adipose tissue expansion is not required for maintenance of whole-body glucose homeostasis or the metabolic response to BAM15-mediated mitochondrial uncoupling.

In summary, we demonstrate that BAM15 is an orally available mitochondrial protonophore that increases energy expenditure to protect against diet-induced obesity and improves glucose homeostasis. The effect of BAM15 on energy expenditure was independent of changes in body temperature and food intake and did not result in the histopathological features present in previous generation protonophores. Taken together, pharmacologic uncoupling by BAM15 or functional derivatives warrants more extensive evaluation for therapeutic potential in the treatment of obesity-related diseases.

### Limitations

BAM15 is a highly lipophilic compound, and as a result, it is not suitable for chronic injection *in vivo*. This limitation was overcome by incorporating BAM15 directly into the diet. As a result, the time and dosing of each animal varied by *ad libitum* food intake. The use of male C57BL/6J mice did not allow for consideration of sex as a biological variable in response to BAM15 at this time.

## Materials and Methods

### Contact for reagent and resource sharing

Further information and requests for resources and reagents should be directed to and will be fulfilled by the Lead Contact, John P. Kirwan (john.kirwan@pbrc.edu).

### Experimental models

#### Tissue culture

C2C12 cells (ATCC, Manassas, VA, USA; passages 4–8) were grown in high-glucose DMEM (4.5 g/l D-glucose) supplemented with 10% fetal bovine serum (FBS), 1% penicillin–streptomycin (100 U/ml), and 0.2% amphotericin B. Prior to 80% confluence, C2C12 cells were plated in accordance with the experimental design. Upon confluence, growth media were removed and replaced with medium to induce myotube differentiation containing high-glucose DMEM supplemented with 2% horse serum, 1% penicillin–streptomycin, and 0.2% amphotericin B. Differentiation medium was replaced every other day for a 5-day period.

AML12 hepatocytes were purchased from ATCC (CRL-2254) and expanded in DMEM:F12 (1:1 mix) supplemented with 10% fetal bovine serum, 10 μg/ml insulin, 5.5 μg/ml transferrin, 5 ng/ml selenium, 40 ng/ml dexamethasone, 1% penicillin–streptomycin, and 0.2% amphotericin B. After reaching ~75% confluence, cells were trypsinized and seeded into a Seahorse XFe24 culture plate (Agilent) at a seeding density of $1.0 \times 10^4$ cells per well.

3T3-L1 preadipocytes were a generous gift from Dr. Elizabeth Floyd at Pennington Biomedical Research Center. Cells were

expanded in DMEM supplemented with 10% bovine calf serum, 1% penicillin–streptomycin, and 0.2% amphotericin B. Upon reaching confluence, cells were trypsinized and seeded into a 0.1% gelatin-coated XFe24 culture plate (Agilent).

## Animal studies

Four-week-old male C57BL/6J mice were purchased at weaning from the Jackson Laboratory (Stock #000664; Bar Harbor, ME, USA). All mice were single-housed and maintained in a conventional animal facility at 21–22°C at a relative humidity of $50 \pm 10\%$ on a 12-h light:dark cycle from time of weaning. All mice were fed a diet consisting of 60% kcal from fat, 20% kcal from protein, and 20% kcal from carbohydrate (VHFD; cat #D12492, Research Diets, New Brunswick, NJ, USA). Animals randomized to BAM15 were fed a diet consisting of 60% kcal from fat, 20% kcal from protein, and 20% kcal from carbohydrate with the addition of 0.1% w/w BAM15 (Research Diets). Each animal was given *ad libitum* access to food (maintaining approximately 50 g of food in the hopper) and water unless indicated. Prior to treatment allocation or experimentation, each animal was weighed twice weekly and body temperature was recorded via infrared thermometry during weighing (Fluke 572-2 Infrared Thermometer; Fluke, Everett, WA, USA). All experiments and procedures involving animals were approved by the Institutional Animal Care and Use Committee of Pennington Biomedical Research Center (permit number 1048).

## Method details

### In vitro cell studies

#### Short Hairpin RNA (shRNA) targeting AMPK

A cell line expressing shRNA to AMPK (shAMPK) was generated by transfecting MISSION® TRC shRNA clone using TransIT®-2020 (Mirus Bio, Madison, WI, USA) according to manufacturer's protocol. In brief, C2C12 cells were transfected with a MISSION AMPK shRNA plasmid (Clone ID: NM_178143.2-1823s21c1, Sigma) or an empty vector shRNA (EV) construct. After 24 h, the media were replaced with DMEM and shAMPK-expressing cells were selected with puromycin (1.5 μg/ml) for 10 days. Knockdown efficiency was confirmed by Western blot against an anti-AMPK antibody. Cells were then grown and differentiated as described above.

#### Caspase 3/7 activity

Caspase 3/7 activation was determined by commercially available luminescent luciferase assay (Promega). C2C12 myoblasts were seeded at 20,000 cells/well in a 96 black walled well plate. After 5 days of differentiation, media were removed and cells were treated with varying concentrations of BAM15, DNP, or FCCP for 16 h in low-glucose DMEM supplemented with 2% bovine serum albumin (BSA). Following treatment, caspase reagent was added directly to the conditioned media and the luminescent intensity was detected on a microplate reader (BioTek Instruments). Data are expressed as the fold induction of caspase 3/7 activity relative to vehicle (0.01% DMSO).

#### NADH dehydrogenase activity

Cell viability was assessed via NADH dehydrogenase activity using a commercially available assay (CCK8; Dojindo, Rockville, MD).

Briefly, fully differentiated C2C12 cells were treated with varying concentrations of BAM15 for 16 h in low-glucose DMEM supplemented with 0.1% BSA. Data are expressed as a percent change relative to vehicle (0.01% DMSO).

### Mitochondrial respiratory kinetics

Oxygen consumption dynamics were assessed by real-time respirometry (Seahorse XFe24; Agilent). C2C12, AML12, and 3T3-L1 cells were seeded at 10,000 cells/well in an XFe 24-well plate (Agilent). Following expansion and differentiation, media were removed and cells incubated for 1 h in XF DMEM (pH 7.4) supplemented with 1 mM pyruvate, 2 mM glutamine, and 10 mM glucose at 37°C, without $CO_2$. Cells were then injected with 1 μM of BAM15, DNP, or FCCP, and the rates of oxygen consumption (OCR) and extracellular acidification (ECAR) were measured over 12 h. Data were normalized to protein content by BCA assay (Thermo Scientific). Maximal OCR was defined as the greatest rate achieved over the 12-h period. Time to maximal OCR was determined by subtracting the time of maximal OCR from baseline injection. The respiratory half-life ($t_{1/2}$) was defined as the time by which the OCR had decayed to 50% of the maximal observed respiration.

### Assessment of mitochondrial respiration in intact cells

Intact cell mitochondrial function was assessed by real-time respirometry (Seahorse XFe24; Agilent). C2C12 myoblasts were seeded at 10,000 cells/well in an XFe 24-well plate (Agilent). After 4 days of differentiation, myotubes were treated with varying concentrations of BAM15 for 16 h. Following treatment, media were removed and cells were incubated for 1 h in XF DMEM (pH 7.4) supplemented with 1 mM pyruvate, 2 mM glutamine, and 10 mM glucose at 37°C, without $CO_2$. Cells were then serially injected with 1 μM oligomycin, 1 μM FCCP, and 0.5 μM of rotenone and antimycin A. Components of mitochondrial function were calculated as described previously (Brand & Nicholls, 2011). Data were normalized to protein content by BCA assay (Thermo Scientific).

### Assessment of OXPHOS and ETC. capacity in permeabilized cells

OXPHOS and ETC. capacity was determined from C2C12 myotubes as described previously (Ye & Hoppel, 2013). Briefly, cells were plated in 10-cm dishes, grown to confluence, and differentiated for 4 days as described above. After 4 days of differentiation, cells were treated with a vehicle (0.01% DMSO) or 20 μM BAM15 in differentiation medium for 16 h. After treatment, cells were removed with 1 ml trypsin-EDTA, transferred into conical tubes containing Hanks buffered saline solution (HBSS), and centrifuged at 350 ×*g* for 5 min at 25°C. The cell pellet was then resuspended in MiR05 medium (mitochondria respiration medium: 110 mM sucrose, 60 mM potassium lactobionate, 0.5 mM EGTA, 3 mM, $MgCl_2 \cdot 6H_2O$, 20 mM taurine, 10 mM $KH_2PO_4$, 20 mM HEPES, and 2 mg/ml BSA, pH = 7.1). A 2 ml suspension containing 500K cells/ml was added into each chamber of an O2K system (OROBOROS). OXPHOS and ETC. capacity was measured using the following concentrations of substrates, uncouplers, and inhibitors: malate (2 mM), pyruvate (2.5 mM), ADP (2.5 mM), glutamate (10 mM), succinate (10 mM), tetramethyl-p-phenylenediamine (TMPD, 0.5 μM), ascorbate (2 mM), carbonylcyanide-p-trifluoromethoxyphenylhydrazone (FCCP, 0.5 μM increment), rotenone (75 nM), antimycin A (125 nM), and sodium azide (200 mM).

## Citrate synthase activity

Citrate synthase activity was determined using a commercially available colorimetric assay (Sigma). Briefly, C2C12 myocytes were treated with a vehicle or 20 μM BAM15 for 16 h. After treatment, cells were washed twice with ice-cold PBS and incubated at 4°C in lysis buffer (Sigma) containing protease inhibitor cocktail (Sigma) for 15 min with gentle shaking. Lysed cells were centrifuged at 20,000 ×g for 15 min to pellet cell debris. The supernatant was then transferred to a fresh tube, and protein content was assessed by BCA assay (Thermo Scientific). 20 μg of protein lysate suspended in 1× assay buffer containing 30 mM acetyl-CoA and 10 mM DTNB was plated in triplicate on a 96-well plate. Absorbance was then measured on a plate reader set to kinetic mode (412 nm, 1.5-min duration, 10-s intervals) before and after the addition of 10 mM oxaloacetate.

## mtDNA

Total DNA was extracted using commercially available reagents (DNeasy Blood and Tissue Kit; Qiagen). RT–qPCR was then performed using Power SYBR Green (Thermo Fisher Scientific) with primers directed against the mitochondrial encoded cytochrome c oxidase subunit II (Cox2) and the nuclear-encoded 18S (Yoon et al, 2010). Primer sequences are found in Appendix Table S1.

## Immunoblotting

C2C12 cells were harvested and lysed for immunoblotting as described previously (Nieuwoudt et al, 2017). Briefly, following treatment, cells were washed twice with ice-cold PBS before addition of lysis buffer (Invitrogen) containing protease inhibitor cocktail (Sigma), 5 mM phenylmethylsulfonyl fluoride (Sigma), and Phos-STOP (Roche Applied Sciences). Lysates were solubilized in Laemmli sample buffer containing 2% β-mercaptoethanol, boiled for 5 min, loaded into 4–12% Tris-Glycine gels, and separated via sodium dodecyl sulfate–polyacrylamide gel electrophoresis at 125 volts for 1.5 h (Invitrogen). Gels were transferred to polyvinylidene fluoride membranes (Bio-Rad) and blocked with 5% BSA in Tris-buffered saline with 0.1% Tween-20 (TBST) for 1 h. Membranes were then incubated overnight with antibodies against pAS160$^{T642}$ (Cell Signaling), AS160 (Millipore), pAMPK$^{T172}$ (Cell Signaling), AMPK (Cell Signaling), pAKT$^{T308}$ (Cell Signaling), AKT (Cell Signaling), GLUT4 (Sigma-Aldrich), Na$^+$K$^+$ATPase (Cell Signaling), or HSC70 (Santa Cruz) diluted 1:1,000 in a 5% BSA solution. Membranes were then washed with TBST and incubated with species-specific horseradish peroxidase-conjugated secondary antibodies (GE Healthcare) diluted 1:10,000 in 5% BSA solution. Immunoreactive proteins were visualized by enhanced chemiluminescence reagent (GE Healthcare) and quantified by densitometric analysis using ImageJ. All antibodies were internally validated prior to use. All bands displayed were subjected to quantification. Values were then expressed as fold induction relative to vehicle normalized to the endogenous control.

## GLUT4 translocation

Plasma membrane GLUT4 was determined by subcellular fractionation as described previously (Malin et al, 2013). Briefly, following 16-h treatment with either vehicle or BAM15 (20 μM), cells were washed three times with ice-cold PBS (pH 7.4). Cells were then scraped and spun at 700 g for 10 min at 4°C. The resulting supernatant was removed, and the pellet was resuspended in HES buffer (250 mM sucrose, 20 mM HEPES, 1 mM EDTA, 1x protease inhibitor cocktail). The cell suspension was then Dounce-homogenized for 30 strokes and centrifuged at 760 ×g for 5 min to remove nuclei and unbroken cells. The resulting supernatant was collected and centrifuged at 31,000 ×g for 60 min to pellet the crude plasma fraction (CPM). The resulting supernatant was collected, transferred to a fresh tube, and spun at 190,000 ×g for 60 min to pellet the light microsomal fraction (LM). Both fractions were resuspended in HES buffer, assayed for total protein content (BCA), and stored at −20°C until time of assay.

## [3-$^3$H]Glucose uptake

Cellular glucose uptake was measured in C2C12 myocytes as described previously with minor modifications (Nieuwoudt et al, 2017). Briefly, following overnight treatment with vehicle or BAM15, cells were incubated in serum-free low-glucose DMEM supplemented with 1% bovine serum albumin for 4 h prior to insulin stimulation. Cells were then stimulated with 1 μM insulin (Novolin-R 100, Novo Nordisk, Plainsboro, NJ) in Krebs–Ringers–HEPES (KRH) buffer (20 mM HEPES, 136 mM NaCl, 4.7 mM KCl, 1.25 mM MgSO$_4$·7H$_2$O, 1.25 mM CaCl$_2$, pH 7.40) for 30 min. Insulin stimulation medium was then removed and replaced with KRH containing 1 μCi/ml [$^3$H]-2-deoxy-D-glucose (Perkin Elmer, Waltham, MA) and 2 mM 2-deoxy-D-glucose. After 10 min of radiolabeled glucose uptake, cells were washed with ice-cold PBS three times and then lysed with 0.1% sodium dodecyl sulfate (SDS) in ddH$_2$O for 30 min with gentle shaking. Lysates were then used for protein quantification (BCA) and scintillation counting in Ultima Gold scintillation fluid (Perkin Elmer). Glucose uptake was calculated relative to vehicle normalized to μg protein.

## In vitro [1-$^{14}$C]palmitate oxidation

Mitochondrial fatty acid oxidation was assessed as previously described with minor modifications (23-26). Following overnight treatment with vehicle or BAM15, cells were incubated in DMEM supplemented with 12.5 mM HEPES, 1 mM L-carnitine, and 100 μM [1-$^{14}$C]-palmitate (Perkin Elmer, Waltham, MA) with either no substrate or glucose (5 mM) and pyruvate (1 mM) for 2 h at 37°C. After incubation, cells were placed on ice and 400 μl of the conditioned media was transferred to borosilicate tubes. Trap tubes containing 200 μl of 1N NaOH were placed in the borosilicate tubes, rubber stoppers were used to seal the borosilicate tubes, and 100 μl of 70% perchloric acid was added. Following a 1-h incubation at room temperature, contents of the NaOH tubes were transferred to scintillation vials, 4 ml of Uniscint BD scintillation cocktail was added, and radiolabel incorporation into $^{14}$CO$_2$ was detected on a scintillation counter. The acidified incubation media were collected and used to detect acid-soluble metabolites. After an overnight incubation, samples were centrifuged at 14,000 ×g for 10 min at 4°C, and an aliquot of the supernatant was removed for scintillation counting as above. Following transfer of the reaction media to the borosilicate tubes, cells were washed with PBS (x3), lysed with 0.1% SDS, and collected after 20 min of gentle agitation at room temperature. All data were normalized to protein content (BCA).

## RNA sequencing

C2C12 myoblasts were grown to confluence and then differentiated for 5 days. Cells were then treated with a vehicle or 20 μM BAM15

for 16 h. After treatment, cells were lysed and total RNA was extracted using RNeasy Mini Kit (Qiagen, Hilden, Germany). RNA was quantified using a NanoDrop and normalized to 200 ng/μl in nuclease-free water. RNA integrity was assessed using an Agilent Bioanalyzer 2100. Libraries were constructed and sequenced using Lexogen QuantSeq. Briefly, library generation was performed using an oligodT primer, and double-stranded cDNA was purified with magnetic beads. Libraries were amplified using PCR, and transcripts were forward-sequenced at 75 bp using NextSeq 500 (Illumina). BlueBee software was used to analyze alignment, and DESeq2 was used for differential expression analysis. Pathway enrichment was analyzed using Ingenuity Pathway Analysis software. Differentially regulated mRNA transcripts were filtered based on the following criteria: $q < 0.01$, base mean > 50, and fold change > 2.

## Animal studies

### BAM15 diet study 1
10-week-old mice ($N = 32$) were randomized 1:1 by a blinded biostatistician according to baseline body weight to 3 weeks of HFD (CTRL) or HFD + BAM15 (0.1% w/w BAM15 in HFD). Body temperature, body weight, and food intake were measured daily. Body composition was measured immediately prior to and after 3 weeks of treatment. Following 3 weeks of treatment, mice were euthanized in their home cage by $CO_2$ inhalation.

### BAM15 diet study 2
10-week-old mice ($N = 24$) were randomized 1:1:1 by a blinded biostatistician according to baseline body weight to 2 weeks of HFD (CTRL), HFD + BAM15 (0.1% w/w BAM15 in HFD), or calorie restriction (CR) to match the body weight achieved in BAM15-treated animals. Body temperature, body weight, and food intake were measured daily. Body composition was measured immediately prior to and after 2 weeks of treatment. Following 2 weeks of treatment, mice were euthanized in their home cage by $CO_2$ inhalation.

### BAM15 acute injection study
10-week-old mice ($N = 16$) were randomized 1:1 by a blinded biostatistician according to baseline body weight to a single intraperitoneal injection of a vehicle solution (5% DMSO in corn oil), 0.1 mg/kg, 0.5 mg/kg, or 1 mg/kg BAM15. Body temperature was assessed immediately prior to, after, and 1 h following injection.

### BAM15 blood and tissue enrichment study
10-week-old mice were switched from HFD to HFD + BAM15 (0.1% w/w BAM15 in HFD). 3 days following the dietary switch, animals were serially harvested every 6 h beginning at 7:00 am over a 24-h period. Serum, gastrocnemius, heart, rpWAT, iWAT, BAT, gWAT, testes, spleen, liver, brain, and kidney were collected at each time point, flash-frozen, and then assessed for BAM15 concentrations via LC-MS/MS.

### BAM15 thermal imaging study
Drug naive male C57BL/6J mice ($N = 3$) on HFD were acclimated to experiment procedure room for 1 h prior to image acquisition. Tail temperatures were recorded on using FLIR

T430sc thermal camera with a 25 degree lens (FLIR Systems Inc., Wilsonville, Oregon, USA). After baseline readings, the diet was switched from HFD to HFD+BAM15, and images were captured over a 24-h period. The camera was positioned approximately 1 m above the cage, and each image was recorded from the same camera height above the cage. Images were analyzed by a blinded technician using FLIR Tools software by selecting area of interest to include the base of the tail and extending approximately 15 mm distally.

### Body temperature
Body temperature was recorded via infrared thermometry during weighing (Fluke 572-2 Infrared Thermometer; Fluke, Everett, WA, USA). The infrared thermometer was calibrated weekly against a rectal probe temperature. Data are reported as the mean of two replicate measures with a CV < 1%.

### Food intake
Food intake was measured to the nearest 0.1 g daily between 09:00 and 11:00 throughout the duration of the study as described previously (Hao *et al*, 2016). Briefly, daily intake was calculated by subtracting the amount of diet recovered from the hopper weight, corrected for spillage found under the grid floor.

### Body composition
Body composition was assessed before and 3 weeks after treatment via nuclear magnetic resonance using LF110 BCA Analyzer (Bruker Corporation, Billerica, MA, USA) as described previously (Halldorsdottir *et al*, 2009). Briefly, at ~07:00, animals were removed from their cage and weighed. Animals were then placed in a restrainer and inserted into the NMR for approximately 2 min. Fat and fat-free mass were determined by calibration with internal standards according to manufacturer's instructions.

### Quantitative determination of BAM15
Frozen tissues were powdered in a tissue pulverizer (Cell-Crusher, USA) cooled by liquid nitrogen. Powdered tissue samples were homogenized in 90% (v/v) acetonitrile (Sigma-Aldrich, 34851, Australia) and 10% (v/v) methanol (Sigma-Aldrich, Australia) using a motorized pellet pestle homogenizer (Sigma-Aldrich, Australia). Homogenate was centrifuged (800 ×g for 10 min) and supernatant collected. Serum was collected from whole blood by centrifugation. BAM15 was extracted by adding tissue homogenate supernatant/serum (1:9) to a solution of 90% (v/v) acetonitrile and 10% (v/v) methanol. The solution was briefly vortexed and then centrifuged (18,000 ×g for 10 min). Supernatant was collected in auto-sampler vials (Thermo Fisher Scientific, Australia) for mass spectrometry. Standards were prepared by spiking known concentrations (0.1, 1, 10, and 100 ng) of BAM15 into untreated tissue or plasma samples prior to extraction. Liquid chromatography–tandem mass spectrometry was performed on a Shimadzu Prominence LCMS-8030 (Shimadzu, Japan). Chromatographic separation was achieved using an ACUITY UPLC BEH, C18 column (Waters, USA). Mobile phase A consisted of 0.1% v/v formic acid (Sigma-Aldrich, Australia) in HPLC-grade water. Mobile phase B consisted of 0.1% v/v formic acid in acetonitrile. The analyte was eluted with a gradient of 5–80% mobile phase B at a flow rate of

0.4 ml/min with 10 µl injection volume electrosprayed into the mass spectrometer. ESI was performed in positive mode. Primary transition of $m/z$ 341 > 162 and secondary transition of $m/z$ 341 > 137 with 6-min retention time were used to identify BAM15. Quantification was determined by measuring peak areas using Shimadzu LabSolutions Software on the instrument. Concentrations of test samples were interpolated from a standard curve derived from the intensity values of standards. Quantified values were normalized to tissue weight.

### Metabolic chamber experiment

Whole-body energy expenditure, oxygen consumption, carbon dioxide production, body weight, and physical activity were continuously monitored for a 7-day period before and 3 weeks after treatment in a mouse metabolic chamber (Sable Systems) as described previously (Burke et al, 2018). Briefly, animals were acclimated to the chamber for 7 days prior to data collection by placement into training cages identical to the metabolic chamber cage. Ad libitum access to food and water was continued while in the training and chamber cages. Average daily and cumulative data were calculated over the final 5 days in the chamber. Locomotor activity was determined by calculating the sum of all detectable motion (> 1 cm/s along X-, Y-, or Z-axis) over the continuous monitoring period.

### Intraperitoneal glucose tolerance test (IPGTT)

Glycemic control was assessed by intraperitoneal injection and subsequent measures of glucose as described previously (Jorgensen et al, 2017). Briefly, after an overnight fast, the animal was restrained and the tail cleaned using a 70% isopropanol wipe. A lancet was then used to create a nick in the tail. After a baseline blood glucose reading (Breeze2 Glucometer, Ascensia Diabetes Care, Parsippany, NJ, USA), a 2 g/kg dose of D-glucose in warmed saline was injected into the peritoneum. Using the same glucometer, blood glucose measurements were taken at 15, 30, 60, 90, and 120 min after injection by reopening the scab created by the initial lancet nick. Area under the curve glucose was determined using the trapezoidal rule (Wolever, 2004). Whole blood was additionally collected at 0 and 120 min followed by centrifugation at 3,000 ×g for 5 min and collection of plasma for assessment of insulin where denoted.

### Determination of fecal lipid content

Fecal lipid content was determined as described previously with minor modifications (Kim & Hoppel, 2013; Kraus et al, 2015). Approximately 250 mg of feces was collected from each animal cage 24 h after a cage change-out. Feces were then hydrated in 100 mM potassium phosphate (pH 7.4) and thoroughly homogenized using a handheld motorized pellet pestle (Kimble-Chase, Vineland, NJ, USA). The fecal homogenate was then transferred to a fresh tube to which 5 ml of a 2:1 v/v chloroform:methanol mixture was added. The tubes were then vortexed vigorously for 2 min and then centrifuged at 2,000 ×g for 10 min. The aqueous layer was removed by aspiration, and the organic layer was removed by puncturing the bottom of the tube with a 16G needle and allowing the chloroform layer to drain to a new tube. Samples were dried for 3–4 days, and the remaining lipids were weighed. Data are shown as percent of feces that is lipid.

### Hematoxylin and eosin and trichrome staining

Tissue sections were collected at necropsy from the following tissues: brain, tongue, heart, lungs, kidney, spleen, liver, stomach, esophagus, small intestine, large intestine, rectum, iWAT, gWAT, retroperitoneal white adipose tissue (rpWAT), mixed gastrocnemius skeletal muscle, soleus skeletal muscle, pancreas, seminal vesicle, and testes. Tissues were grossed to size and fixed in 10% neutral-buffered formalin for 72 h, changing the fixative every 24 h. Tissues were then paraffin-embedded, sectioned to a width of 5 µm, and fixed to a glass slide. Slides were then stained for hematoxylin and eosin as described previously (Fischer et al, 2008). In cases with fibrosis, slides were stained with Trichrome via commercially available reagents (Poly Scientific R&D, Bay Shore, NY). Pathological features were determined by a blinded, board-certified veterinary pathologist. Quantitative analysis of lipid content from liver and adipose tissue sections was performed using Adiposoft package in Image J.

### Immunohistochemistry staining

Mouse pancreas was fixed in 10% neutral-buffered formalin for 72 h, the tissue was then embedded in paraffin, and cut into 5-µm sections. Immunohistochemistry was performed on a Leica Bond-Max System (Leica Biosystems, Melbourne, Australia) using the Bond Polymer Refine Detection Kit (Leica Biosystems, Melbourne, Australia). Anti-insulin (1:500, Invitrogen, Carlsbad, CA) antibodies were used followed by 30 min with HRP-conjugated secondary antibody (1:500, MilliporeSigma, Burlington, MA). Anti-PDX-1 (1:500, Abcam, Cambridge, MA) antibodies were used for overnight incubation at 4°C. 3,3'-Diaminobenzidine (DAB) was used for detection. Hamamatsu NanoZoomer digital slide scanner (Hamamatsu, Bridgewater, NJ) was used for imaging. For the determination of insulin-positive area, four mice per group were used, and data were analyzed as the ratio of insulin-positive area to total pancreatic tissue area. For PDX-1, data were expressed as the ratio of nucleus PDX-1-positive area to total islet area. The sections were analyzed and quantified using Fiji Image J software (Schindelin et al, 2012).

### Ex vivo [1-$^{14}$C]Palmitate oxidation

Mixed gastrocnemius muscle collected at the time of necropsy was used fresh to prepare homogenates using a procedure that yields > 75% intact mitochondria (Fuller et al, 2019). Homogenates were incubated in the presence of [1-$^{14}$C]-palmitate ± 1 mM unlabeled pyruvate to assess mitochondrial substrate selection as described previously (Noland et al, 2007a,b). Complete palmitate oxidation was measured as liberation of $^{14}CO_2$ following a 30 min reaction, while [$^{14}$C] radiolabel incorporation into acid-soluble metabolites served as an index of incomplete fatty acid oxidation. Standard liquid scintillation counting was used to measure presence of radiolabel, and results are expressed as nmol of oxidized substrate per gram tissue per hour.

### Assessment of OXPHOS and ETC. capacity in permeabilized muscle fibers

OXPHOS and ETC. capacity was determined ex vivo from permeabilized mixed gastrocnemius fibers as described previously (Ye & Hoppel, 2013). During necropsy, 20–25 mg of mixed gastrocnemius muscle was collected and immediately placed into BIOPS (50 mM K$^+$-MES, 20 mM taurine, 0.5 mM dithiothreitol, 6.56 mM MgCl$_2$,

5.77 mM ATP, 15 mM phosphocreatine, 20 mM imidazole, pH 7.1, adjusted with 5 N KOH at 0°C, 10 mM Ca–EGTA buffer, 2.77 mM $CaK_2EGTA$ + 7.23 mM $K_2EGTA$, and 0.1 mM free calcium) solution. The muscle bundles were then mechanically separated under a dissection microscope, placed into fresh BIOPS containing saponin (5 mg/ml), and gently agitated at 4°C for 20 min. The fibers were then transferred to a mitochondrial respiration medium (MiR05; 110 mM sucrose, 60 mM $K^+$-lactobionate, 0.5 mM EGTA, 3 mM $MgCl_2$, 20 mM taurine, 10 mM $KH_2PO_4$, 20 mM HEPES adjusted to pH 7.1 with KOH at 37°C, and 1 g/l de-fatted BSA), blotted on filter paper, and weighed. 2–5 mg of permeabilized fiber bundles were transferred into the oxygraph chamber containing 2 ml of MiR05 until background respiration was stable. OXPHOS and ETC. capacity was measured using the following concentrations of substrates, uncouplers, and inhibitors: malate (2 mM), pyruvate (2.5 mM), ADP (2.5 mM), glutamate (10 mM), succinate (10 mM), palmitoylcarnitine (10 μM), duroquinol (0.5 mM), tetramethyl-p-phenylenediamine (TMPD, 0.5 μM), ascorbate (2 mM), carbonylcyanide-p-trifluoromethoxyphenylhydrazone (FCCP, 0.5 μM increment), rotenone (75 nM), antimycin A (125 nM), and sodium azide (200 mM).

### Biochemical measurements

Blood samples were collected by cardiac puncture in animals under terminal anesthesia into a minivette capillary tube coated with $K_3$-EDTA. Plasma was isolated by centrifugation at 2,000 $\times g$ for 20 min at 4°C. Samples were aliquoted and stored at −80°C until assay. Plasma insulin (Mercodia), FGF21 (R&D Systems), GDF15 (R&D Systems), leptin (R&D Systems), and adiponectin (R&D Systems) concentrations were measured via enzyme-linked immunosorbent assay (ELISA) per manufacturer's instructions. Non-esterified fatty acids (NEFA; Abcam) and beta-hydroxybutyrate (BHB; Cayman Chemical) concentrations were measured via colorimetric assay per manufacturer's instructions.

### Tissue homogenization and immunoblotting

Mixed gastrocnemius, heart, liver, and iWAT homogenates were prepared in ice-cold cell extraction buffer (Invitrogen) with protease inhibitor cocktail, 5 mM phenylmethylsulfonyl fluoride (Sigma), and Phos-STOP (Roche Applied Sciences, Indianapolis, IN) as described previously (Axelrod et al, 2018). Immunoblotting performed as described above against the following primary antibodies: $pAMPK^{T172}$ (Cell Signaling), AMPK (Cell Signaling), $pACC^{S79}$ (Cell Signaling), ACC (Cell Signaling), vinculin (Cell Signaling), or HSC70 (Santa Cruz) diluted 1:1000 in a 5% BSA solution. Values are expressed as fold induction relative to CTRL animals normalized to the endogenous control.

### Ex vivo lipolysis

Adipose tissue lipolytic rates were measured as described previously (Schweiger et al, 2014). Briefly, iWAT and gWAT depots were obtained at the time of necropsy, sectioned into 20–30 mg pieces, and placed in an incubation medium composed of phenol red-free DMEM containing 0.1% glucose and 2% de-fatted bovine serum albumin for 5 h. After incubation, the media were removed and assayed for free glycerol using a commercially available colorimetric assay (Sigma). Data are expressed as mg glycerol/ml incubation solution/mg tissue/hr.

### RNA isolation/cDNA synthesis/real-time PCR

Total RNA was extracted from iWAT using RNeasy Isolation Kit (Qiagen, Germantown, MD) as per manufacturer's instructions. The RNA concentration was determined on a microvolume spectrophotometer (NanoDrop 8000; Thermo Fisher). 1 μg of total RNA was treated using an iScript cDNA Synthesis Kit (Bio-Rad, Hercules, CA) in a PX2 Thermal Cycler (Thermo Fisher Scientific, Waltham, MA) and converted to cDNA by reverse transcription. Quantitative RT–PCR was carried out with Power SYBR Green PCR Master Mix (Bio-Rad, Hercules, CA) and different primers in QuantStudio 5 real-time PCR system (Applied Biosystems, Foster City, CA). The primer sequences are shown in Appendix Table S1. All reactions were carried out in quadruplicate. Ct values were obtained and normalized to Gapdh expression. Relative fold expressions from qRT–PCR were determined using the comparative $\Delta\Delta C_t$ method.

### Transmission electron microscopy (TEM)

Mixed gastrocnemius sections were collected at the time of necropsy and assessed for mitochondrial ultrastructure and content as described previously (Fujioka et al, 2014a,b). Briefly, small pieces of tissue were fixed by immersion in quarter strength Karnovsky's fixative solution (Electron Microscopy Sciences, Hatfield, PA, USA). The specimens were thoroughly rinsed in 0.1 M phosphate buffer, pH 7.4, and then postfixed for 2 h in an unbuffered 1:1 mixture of 2% osmium tetroxide and 3% potassium ferricyanide. After rinsing with distilled water, the specimens were soaked overnight in an acidified solution of 0.25% uranyl acetate. After another rinse in distilled water, they were dehydrated in ascending concentrations of ethanol, passed through propylene oxide, and embedded in Embed-812 resin mixture (Electron Microscopy Sciences, Hatfield, PA). Thin sections were sequentially stained with acidified uranyl acetate followed by a modification of Sato's triple lead stain and examined in a FEI Tecnai Spirit (T12) with a Gatan US4000 4kx4k CCD.

### Quantification and statistical analysis

Data are reported as mean ± standard error of the mean (SEM) unless otherwise denoted in the figure legend. Prism 8 software (GraphPad, San Diego) was used for statistical testing. Ingenuity (Qiagen) was used to conduct untargeted pathway analysis using the following parameters: q < 0.01 and log fold induction > 2. Differential analysis of RNA read count data was performed using DESeq2 software, which models read counts as a negative binomial distribution and uses an empirical Bayes shrinkage-based method to estimate signal dispersion and fold changes as described previously (Ghosh et al, 2019). Gene expression signals were logarithmically transformed (to base 2) for all downstream analyses (the lowest expression value being set to 1 for this purpose). Statistical analysis of RNA sequencing was performed using R (Vienna, Austria). Animals were randomized either 1:1 or 1:1:1 by a blinded biostatistician to treatment based on baseline body weight. Histological evaluations were qualitatively assessed by a blinded veterinary pathologist. Body weight, body temperature, food intake, and IPGTT and IPITT were not blinded to the experimenter but were blinded to the biostatistician. Statistical procedures from individual experiments are detailed in the respective figure legends. Normality of the models was assessed by Kolmogorov–Smirnov and D'Agostino–

**The paper explained**

**Problem**
Obesity is a growing epidemic and a leading cause of preventable death worldwide. There is a critical, unmet need for medications that effectively treat the multifactorial causes and consequences of obesity.

**Results**
The present study demonstrates that (i) BAM15, a small molecule mitochondrial protonophore, exhibits superior respiratory kinetics and tolerability to previous generation drugs of the same class; (ii) sustained uptake of glucose and fatty acids by BAM15 is AMPK-dependent; (iii) BAM15 is orally available, selective to lipophilic tissues, prevents dietary weight gain, and improves glucose homeostasis by increasing energy expenditure in obese C57BL/6J mice; and (iv) AMPK and ACC are activated in iWAT following treatment with BAM15 which contributes to reduced adipocyte size and expression of genes regulating *de novo* lipogenesis.

**Impact**
These data provide a mechanism of action and demonstrate pre-clinical efficacy for BAM15, an approach for the treatment of obesity and related metabolic disorders.

Pearson test where appropriate. Significance was accepted as $P < 0.05$.

## Data availability

The RNAseq datasets produced in this study are available at Gene Expression Omnibus (GEO) under the accession number GSE138790 (https://www.ncbi.nlm.nih.gov/geo/query/acc.cgi?acc = GSE138790).

**Expanded View** for this article is available online.

## Acknowledgements

We thank the Cell Biology and Bio-imaging Core Facility for tissue sample processing for histology. We thank the Genomics Core for conducting RNA sequencing. We thank the Comparative Biology Core staff for animal husbandry. This work used core facilities that are supported in part by COBRE (NIH 8 P20 GM103528-07) and NORC (NIH 1P30-DK072476) center grants from the National Institutes of Health. This research was supported in part by National Institutes of Health grants DK108089 (JPK), U54GM104940 (JPK), DK103860 (RCN), and DK115749 (KS), as well as National Health and Medical Research Council grant GNT1163903 (KLH).

## Author contributions

Conceptualization, CLA, WTK, CF, and JPK; Investigation, CLA, WTK, GD, RCN, JH, MH, WSD, ERMZ, SJA, KLH, IL, KS, HD, ES, SN, KF, KP, HF, JTM, AM, RB, and CLH; Writing—Original Draft, CLA; Writing—Reviewing & Editing, all authors; Funding Acquisition, RCN, KS, and JPK; Supervision, CLA and JPK.

## Conflict of interest

K.L.H. is an equity holder in Continuum Biosciences, a company that has commercial interest in mitochondrial uncouplers as therapeutics. All other authors declare no conflict of interest.

## For more information

(i) Authors webpage: https://www.pbrc.edu/research-and-faculty/faculty/?faculty=5489

(ii) Information for persons with obesity: https://www.pbrc.edu/research-and-faculty/faculty/?faculty=5489

(iii) Information for persons with type 2 diabetes: https://www.diabetes.org/diabetes/type-2

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
