## [Review Process File · EMBO Molecular Medicine]

BAM15 Mediated Mitochondrial Uncoupling Protects Against Obesity and Improves Glycemic Control

Christopher Axelrod, William King, Gangarao Davuluri, Robert Noland, Jacob Hall, Michaela Hull, Wagner Dantas, Elizabeth Zunica, Stephanie Alexopoulos, Kyle Hoehn, Langohr Ingeborg, Krisztian Stadler, Haylee Doyle, Eva Schmidt, Stephan Nieuwoudt, Kelly Fitzgerald, Kathryn Pergola, Hisashi Fujioka, Jacob Mey, Ciaran Fealy, Anny Mulya, Robbie Beyl, Charles Hoppel, and John Kirwan
DOI: 10.15252/emmm.202012088

Corresponding author(s): John Kirwan (john.kirwan@pbrc.edu) , Christopher Axelrod (Christopher.Axelrod@pbrc.edu)

Review Timeline:

Submission Date:	29th Jan 20
Editorial Decision:	14th Feb 20
Revision Received:	20th Apr 20
Editorial Decision:	14th May 20
Revision Received:	15th May 20
Accepted:	15th May 20

Editor: Zeljko Durdevic

Transaction Report:

14th Feb 2020

Dear Dr. Kirwan,

Thank you for the submission of your manuscript to EMBO Molecular Medicine. We have now heard back from the three referees who agreed to evaluate your manuscript. As you will see from the reports below, while referee 3 is supportive of publication, referees 1 and 2 raise a number of concerns that should be addressed in a major round of revision of the present manuscript.

A cross-commenting exercise clarified the major issues that should be addressed: 1. short-term BAM15 in vivo treatment to determine baseline and IPGTT insulin levels, and 2. experiments to determine whether BAM15 alters beta cell function independently of weight. Please note that addressing all the other points raised by the referees as much as possible will be necessary for further considering the manuscript in our journal, and acceptance of the manuscript will entail a second round of review. EMBO Molecular Medicine encourages a single round of revision only and therefore, acceptance or rejection of the manuscript will depend on the completeness of your responses included in the next, final version of the manuscript. For this reason, and to save you from any frustrations in the end, I would strongly advise against returning an incomplete revision.

***** Reviewer's comments *****

Referee #1 (Remarks for Author):

The work by Axelrod et al. investigates the effects of a small molecule protonophore, referred to as BAM15, on cellular respiration in vitro and whole-body energy metabolism in mice. The manuscript is novel, very well prepared and the conclusions are altogether supported by the data. While it might not come as a surprise that a mitochondrial uncoupler increases mitochondrial leak respiration independently of mitochondrial capacity and increases energy expenditure in whole animals, the data nicely show that this compound may have therapeutic potential for obesity-linked diseases.

I have a couple of experimental and/discussion points the authors may want to consider:

Major:

1. The authors point to their body core measurement to conclude that BAM15 does not impact body temperature. However, the dissipated calories have to go somewhere, as clearly energy expenditure is up after the high-fat diet feeding. The mouse is not stupid and has other mechanisms to regulate body temperature. A major, yet underappreciated organ for maintaining body core temperature is the tail. When the mouse is cold, heat loss through the tail is minimized by vasoconstriction and vice versa. So, I hypothesize that while body core temperature is unaltered, BAM15 treatment likely increases heat loss via the tail. This can be measured or discussed.

2. In relation to that, especially in light of the bioavailability of the compound, that the authors somehow neglected to look at brown adipose tissue (BAT) is a little weird. In mice kept at room temperature non-shivering thermogenesis is a major contributor to energy expenditure and most of the effects could be due to increased BAT function. It might seem counterintuitive, as this tissue already expresses Ucp1 as a natural uncoupler, but it nevertheless should be looked at to fill in all

the pieces of the puzzles (e.g. by histology, Ucp1 expression, similarly to iWAT and muscle)

3. Finally, these experiments were done at room temperature but for humans thermoneutrality (ca. 30 degrees Celsius) in mice is a much better comparison. Are the effects of the compound dependent of the housing temperature of the mouse?

Minor:

4. The necessity of some of the long-term BAM15 effects in cells likely reflect a general energy depletion rather than a specific effect of the compound on the AMPK pathway. This should be noted.

5. The title should be edited for better visibility of the study, e.g. Protonophore BAM15 etc.

Referee #2 (Remarks for Author):

In the manuscript entitled « BAM 15 prevents dietary weight gain and improves glycemic control in obese mice », Christopher L Axelrod et al. describe the effect of the compound BAM15, a protonophore targeting mitochondria, in mitochondrial activity, energy expenditure, glucose and lipid metabolism. Using cellular, transcriptomic, in vivo approaches, the authors claim that the BAM15 compound mainly target adipose tissue and gastrocnemius/muscle when injected in C57Bl6J, where it increases AMPK activity, resulting in increased fatty acid oxidation. Altogether, these data are interesting in the fact that BAM15 does not seem to be toxic, while improving glucose and lipid homeostasis in an obesity context.

Although the study is highly interesting, demonstrating that protonophore, in particular BAM15 could be used as a potential new drug against obesity, NASH, ... the data presented are too preliminary at this stage and need important clarifications before being published. Moreover, the general feeling of the reviewer is that it is difficult to follow the results, since most of the figures are not well presented. For example, many figures are not numbered, and each graph/data should be numbered to help the reader to follow the manuscript. Some conclusions are not supported by the data.

Concerning the in vitro part, several controls of experiments should be shown, such as, for example but not limited to, shcontrol in C212 treated with BAM15 (figure 3).

I can understand that the transcriptomic analysis is done on cell lines, but it can really be different from what can happen in mice. I would suggest to do a RNA-seq in tissues of mice treated with the vehicle or BAM15, or at least to confirm some of the genes that were modulated in C2C12. Looking at the pathway analysis, it seems that cholesterol metabolism is the most affected pathway, whereas the focus was done on AMPK. Why the authors did not study the cholesterol pathway in BAM15 treated cells?

Insulin tolerance tests should be performed to analyze the effect of BAM15 on insulin sensitivity. This is key to demonstrate that BAM15 decreases insulin resistance, in a context where beta cell mass is decreased, FGF21, GDF15 are also decreased. In addition, insulin levels should be measured during ipGTT to demonstrate whether improved glucose tolerance is linked to increased insulin secretion in response to a bolus of glucose. This can then be correlated (or not) to IHC analysis of pancreatic sections, which show a clear decrease in insulin staining. Beta cell mass should be measured more precisely. Finally, to demonstrate, as claimed by the authors, that BAM15 is an

AMPK agonist (which is not true, since no CETSA nor in vitro AMPK activity demonstrate that BAM15 binds to AMPK) or at least exerts part of its effect through the activation of AMPK, in adipose tissue, the use of AMPK inhibitor or genetic deficient mice could have been used to unequivocally demonstrate the interaction between BAM15 and AMPK pathways in regulating energy homeostasis.

Referee #3 (Remarks for Author):

In this study Axelrod et al. examine the mitochondrial protonophore BAM15 in vivo and in vitro as an anti-obesity agent. As the authors describe, this drug has already been nicely shown to uncouple in vitro, but the current study extends this work, showing in vivo that it has anti obesity and insulin sensitizing properties in vivo. They are appropriately measured with their conclusions. I congratulate them on this nice study.

Referee #1 (Remarks for Author):

The work by Axelrod et al. investigates the effects of a small molecule protonophore, referred to as BAM15, on cellular respiration in vitro and whole-body energy metabolism in mice. The manuscript is novel, very well prepared and the conclusions are altogether supported by the data. While it might not come as a surprise that a mitochondrial uncoupler increases mitochondrial leak respiration independently of mitochondrial capacity and increases energy expenditure in whole animals, the data nicely show that this compound may have therapeutic potential for obesity-linked diseases. I have a couple of experimental and/discussion points the authors may want to consider.

RESPONSE: We thank the reviewer for their time and interest in our work. We have performed additional experiments and revised the manuscript as denoted below to address the concerns of the reviewer.

Major:

1) The authors point to their body core measurement to conclude that BAM15 does not impact body temperature. However, the dissipated calories have to go somewhere, as clearly energy expenditure is up after the high-fat diet feeding. The mouse is not stupid and has other mechanisms to regulate body temperature. A major, yet underappreciated organ for maintaining body core temperature is the tail. When the mouse is cold, heat loss through the tail is minimized by vasoconstriction and vice versa. So, I hypothesize that while body core temperature is unaltered, BAM15 treatment likely increases heat loss via the tail. This can be measured or discussed.

RESPONSE: We thank the reviewer for their insights on this matter. The data previously included was measured by skin thermometry validated against a rectal probe, which reflects core body temperature. These studies were conducted to determine if BAM15 protected against HFD-induced obesity without induction of hyperthermia, a side effect of other mitochondrial protonophores [1]. To better address mechanisms of heat dissipation, we conducted infrared whole-body thermography before, during, and after oral consumption of BAM15 in 1-year old weight-stable wild-type C57BL/6J mice. This approach allows for direct quantitation of tail heat dissipation as described previously [2]. We observed that BAM15 did not alter tail heat dissipation or body temperature under any conditions (**Response Figure 1**). To confirm that BAM15 was metabolically active, energy expenditure was assessed in real time over the same treatment duration (**Response Figure 1**). BAM15 increased energy expenditure during the dark phase similar to what was observed in our diet-induced obese animals. Though we do not have a conclusive answer as to how BAM15 stimulates mitochondrial respiration and energy expenditure without altering whole-body or tail temperature, Rajagopal *et al.* recently employed intracellular thermometry to measure the effect of uncoupling on temperature flux [3]. The study demonstrated that BAM15-mediated uncoupling (10 μ M BAM15) of neuronal mitochondria produced a maximal intracellular temperature flux of $\sim 8^{\circ}\text{C}$, which normalized within ~ 30 seconds [3]. Notably, the extracellular flux was $\sim 2^{\circ}\text{C}$, and normalized within ~ 10 seconds. As such, it may be that compounds, such as DNP, that lack mitochondrial specificity result in significantly greater extracellular heat production, subsequently resulting in hyperthermia at higher dosing.

Response Figure 1. Effect of BAM15 on Tail Heat Dissipation and Energy Expenditure.

2) In relation to that, especially in light of the bioavailability of the compound, that the authors somehow neglected to look at brown adipose tissue (BAT) is a little weird. In mice kept at room temperature non-shivering thermogenesis is a major contributor to energy expenditure and most of the effects could be due to increased BAT function. It might seem counterintuitive, as this tissue already expresses Ucp1 as a natural uncoupler, but it nevertheless should be looked at to fill in all the pieces of the puzzles (e.g. by histology, Ucp1 expression, similarly to iWAT and muscle)

RESPONSE: We agree with the reviewer that endogenous uncoupling mechanisms, most notably UCP expression, could explain the increase in energy expenditure observed in BAM15 treated animals. To address this, we have additionally stained BAT and iWAT for UCP1 expression from animals treated for 3 weeks with BAM15 as described in the primary manuscript. BAT UCP1 was highly expressed in both control and BAM15 treated animals, but did not differ between groups (**Response Figure 2**). As noted in Figure 5A, although the number of brown adipocytes increased, this was more so a function of reduced size. We additionally stained iWAT for UCP1 to identify beiging of the depot by BAM15, all of which were negative (**Response Figure 3**). UCP1, to our knowledge, has low or no endogenous expression in mammalian skeletal muscle [4], and is only inducible by β_3 -adrenergic receptor agonists [5]. However, the homologues UCP2 and UCP3 are expressed in skeletal muscle. We used our RNA sequencing from murine fibroblasts and observed that UCP2 expression (**Response Figure 4**) was significantly reduced by BAM15. UCP3 was unchanged by treatment. We therefore concluded that neither BAT nor beiging of iWAT contribute to increased energy expenditure following treatment with BAM15. It is our belief that exogenous respiratory uncoupling neutralizes or substitutes for endogenous uncoupling activity by reducing cellular demand for activation, as originally postulated by Goldgof *et al.* [6].

Response Figure 2. BAT UCP1 Expression in CTRL and BAM15-treated animals.

Response Figure 3. iWAT UCP1 Expression in CTRL and BAM15-treated animals.

Response Figure 4. UCP2 Gene Expression in C2C12 cells after 16 hours of Vehicle or BAM15.

3) Finally, these experiments were done at room temperature but for humans thermoneutrality (ca. 30 degrees Celsius) in mice is a much better comparison. Are the effects of the compound dependent of the housing temperature of the mouse?

RESPONSE: We agree with the reviewer and acknowledge that housing at 22°C (human thermoneutrality) instead of 30°C (murine thermoneutrality) could influence weight maintenance and/or loss by exacerbating facultative thermogenesis. We tested phenotypic dependence on thermoneutrality by acclimating C57BL/6J mice to 30°C (range of room: 28-30°C) and subsequently exposing to HFD (CTRL) or HFD+BAM15 (BAM15). Body temperature did not differ between HFD and BAM15-treated animals (**Response Figure 5**). We observed that by day 3, BAM15 treated animals weighed significantly less than CTRL (**Response Figure 5**). Food intake relative to body weight was slightly increased in BAM15 treated animals, which diminished over the treatment period (**Response Figure 5**). Though we fundamentally agree with the reviewer that testing of anti-obesity compounds is more physiologically relevant at 30°C [6], housing temperature did not alter the efficacy and/or availability of BAM15.

Response Figure 5. Change in body temperature, weight, and food intake at thermoneutrality.

Response Figure 5. Change in body temperature, weight, and food intake at thermoneutrality.

Minor:

4) The necessity of some of the long-term BAM15 effects in cells likely reflect a general energy depletion rather than a specific effect of the compound on the AMPK pathway. This should be noted.

RESPONSE: The manuscript has been updated to include discussion of energy depletion as it pertains to activation of AMPK (Lines 382-384).

5) The title should be edited for better visibility of the study, e.g. Protonophore BAM15 etc.

RESPONSE: The manuscript title has been updated to improve visibility (Lines 1-2).

Referee #2 (Remarks for Author):

1) In the manuscript entitled « BAM 15 prevents dietary weight gain and improves glycemic control in obese mice », Christopher L Axelrod et al. describe the effect of the compound BAM15, a protonophore targeting mitochondria, in mitochondrial activity, energy expenditure, glucose and lipid metabolism. Using cellular, transcriptomic, in vivo approaches, the authors claim that the BAM15 compound mainly target adipose tissue and gastrocnemius/muscle when injected in C57Bl6J, where it increases AMPK activity, resulting in increased fatty acid oxidation. Altogether, these data are interesting in the fact that BAM15 does not seem to be toxic, while improving glucose and lipid homeostasis in an obesity context. Although the study is highly interesting, demonstrating that protonophore, in particular BAM15 could be used as a potential new drug against obesity, NASH, ... the data presented are too preliminary at this stage and need important clarifications before being published. Moreover, the general feeling of the reviewer is that it is difficult to follow the results, since most of the figures are not well presented. For example, many figures are not numbered, and each graph/data should be numbered to help the reader to follow the manuscript. Some conclusions are not supported by the data.

RESPONSE: We thank the reviewer for highlighting the potential clinical applications of our findings. We have substantially revised the manuscript to improve readability, including labeling of all individual figures. To address the reviewer's concerns regarding the study conclusions, we performed additional experiments and added controls as described below.

2) Concerning the in vitro part, several controls of experiments should be shown, such as, for example but not limited to, shcontrol in C2C12 treated with BAM15 (figure 3).

RESPONSE: We agree that accounting for plasmid introduction is an important experimental control, and have since included throughout Figure 3 to improve data clarity and conclusiveness. Regarding the remainder of wild type C2C12 experiments, BAM15 treatments were compared to either a vehicle with matched DMSO concentrations (variable based on BAM15 concentration) or equimolar solutions of DNP and/or FCCP. We did not include a non-vehicle (negative) control across experiments as the primary BAM15 treatment (20 μ M BAM15) was dissolved in DMSO (0.01% (v/v) final) at a concentration where negligible cellular effects, such as cytotoxicity and apoptosis, are observed [7].

3) I can understand that the transcriptomic analysis is done on cell lines, but it can really be different from what can happen in mice. I would suggest to do a RNA-seq in tissues of mice treated with the vehicle or BAM15, or at least to confirm some of the genes that were modulated in C2C12.

RESPONSE: We also agree with the reviewer in that cellular transcriptomic profiling can greatly differ from what is observed in tissues. For this work, RNA sequencing from cells was performed prospectively as a means to identify regulatory pathways of interest, not to identify/demonstrate a genetic basis for mechanism of action. Notably, the AMPK subunits were only modestly altered by BAM15 treatment. This would be expected due to the rapid and sustained phosphorylation of AMPK at Thr172 in response to increases in the cellular AMP:ATP ratio. In this case, we would expect that if gene expression of AMPK was altered, the total protein expression, rather than phosphorylation, would be altered. From our cell and tissue experiments, total AMPK protein expression was unaltered by BAM15. Though we have interest in conducting further sequencing of tissues from BAM15 treated animals, tissue was not available at this time for further analysis.

4) Looking at the pathway analysis, it seems that cholesterol metabolism is the most affected pathway, whereas the focus was done on AMPK. Why the authors did not study the cholesterol pathway in BAM15 treated cells?

RESPONSE: The cholesterol synthesis pathway was significantly upregulated by BAM15 treatment in C2C12 cells. Though this is certainly of interest to the investigative team, AMPK-related signaling was chosen for subsequent protein validation due to the well established relationship between bioenergetic efficiency and AMPK. We believed it to be that cholesterol biosynthesis was a required element linking reduced bioenergetic efficiency to improvements in nutrient uptake. Rather, we postulate that it was likely a compensatory mechanism to energy depletion, which was possibly more severe in vitro due to the inability to mobilize circulating nutrients, unlike what would occur in mice.

5) Insulin tolerance tests should be performed to analyze the effect of BAM15 on insulin sensitivity. This is key to demonstrate that BAM15 decreases insulin resistance, in a context where beta cell mass is decreased, FGF21, GDF15 are also decreased. In addition, insulin levels should be measured during ipGTT to demonstrate whether improved glucose tolerance is linked to increased insulin secretion in response to a bolus of glucose. This can then be correlated (or not) to IHC analysis of pancreatic sections, which show a clear decrease in insulin staining.

RESPONSE: We agree with the reviewer that determining the relative contribution of insulin sensitivity, secretion, or both in mediating the metabolic improvements observed after BAM15 treatment is of great importance. Based upon the comments of this reviewer and the cross-reviewing exercise, we performed an additional set of experiments whereby male C57BL/6J mice were randomized to two weeks of HFD (CTRL), HFD+BAM15 (BAM15), or calorie restriction (CR) to match the body weight achieved in BAM15 treated animals. These data are now represented in **Figure 5** of the manuscript. After 2 weeks of treatment, BAM15 and CR animals achieved equally reduced body weight relative to CTRL (**Figure 5A**). CR animals required ~15% restriction of food intake, whereas food intake did not differ between CTRL and BAM15 treated animals (**Figure 5D**). Interestingly, BAM15 reduced fat mass and preserved lean mass to a greater extent than CR (**Figure 5B-C**). We then performed IPGTTs with measurement of both glucose (0,15,30,60,90, and 120 min) and insulin (0 and 120 min). Fasting glucose did not differ between groups (**Figure 5E**). BAM15 animals displayed improved glucose tolerance relative to both CTRL and CR (**Figure 5E-F**). Baseline (PRE) concentrations of insulin did not differ between groups (**Figure 5G**). After the two-week treatment period (POST), BAM15 and CR treated animals had equally reduced fasting insulin concentrations relative to CTRL (**Figure 5G**). Similar to glucose, the 120-minute insulin concentrations in BAM15 treated animals were reduced relative to both CTRL and CR (**Figure 5G**). HOMA-IR was then calculated as a validated index of insulin sensitivity, which was equally reduced in BAM15 and CR relative to CTRL animals (**Figure 5H**), suggesting that the effects of BAM15 on insulin sensitivity are weight dependent, whereas its effect on insulin secretion are additive to weight loss. To further support a direct effect of BAM15 on insulin secretion, INS-1 832/13 pancreatic β -cells were pre-treated for 2 hours with varying concentrations of BAM15 under low and high glucose conditions (**Response Figure 6**). Consistent with our histological, biochemical, and in vivo findings, BAM15 reduced GSIS, which we would expect since tight coupling of electron transport to oxidative phosphorylation is a major driver of insulin secretion. From these experiments we concluded that BAM15 has weight independent effects on glycemic control, which we attributed to greater reductions in fat mass and reduced insulin demand.

Response Figure 6. The Effect of BAM15 on GSIS *in vitro*.

6) Beta cell mass should be measured more precisely.

RESPONSE: We have now included quantitation of beta-cell mass, which was determined by multiplying the ratio of insulin positive to total pancreatic area (%) by pancreatic wet weight measured at the time of necropsy as described previously [8]. These data are displayed in manuscript Figure 6F.

7) Finally, to demonstrate, as claimed by the authors, that BAM15 is an AMPK agonist (which is not true, since no CETSA nor in vitro AMPK activity demonstrate that BAM15 binds to AMPK) or at least exerts part of its effect through the activation of AMPK, in adipose tissue, the use of AMPK inhibitor or genetic deficient mice could have been use to unequivocally demonstrate the interaction between BAM15 and AMPK pathways in regulating energy homeostasis.

RESPONSE: We thank the reviewer for their observations and agree that it is unlikely that BAM15 binds to or directly interacts with AMPK. It is our belief that AMPK functions as a catabolic energy sensor that is activated in response to reduced bioenergetic efficiency as it would occur by other forms of energy depletion, such as starvation or exercise. We have updated language in the manuscript to reflect this notion (Line 382-384). Additionally, we agree with the reviewer that an in vivo experiment where AMPK is inhibited and/or genetically modified would more conclusively demonstrate the role of AMPK in BAM15 mediated improvements in weight regulation and glycemic control. However, it is our belief that such approaches are outside of the scope of this investigation for the following reasons: 1) There are no AMPK-specific pharmacological inhibitors. Dorsomorphin (also known as Compound C) is in our view the most suitable candidate, as it does inhibit AMPK signaling and is somewhat widely used. Dorsomorphin is also highly non-specific, equally if not more potently inhibiting bone morphogenic protein (BMP) and transforming growth factor beta (TGF β) signaling [9, 10]. 2) Given the limitations of currently available pharmacological approaches, genetically deficient mice would be the most suitable approach. Specifically, deletion of AMPK from iWAT or adipose tissue would be required for this approach. To our knowledge, such a mouse is not commercially available and as such would require considerable time to generate, validate, and back-cross over multiple generations. Generating such a mouse, though of great interest to our research and certainly a focus moving forward, would significantly delay the dissemination of the current findings and in our view, does not alter the primary outcomes of the study.

Referee #3 (Remarks for Author):

1) In this study Axelrod et al. examine the mitochondrial protonophore BAM15 in vivo and in vitro as an anti-obesity agent. As the authors describe, this drug has already been nicely shown to uncouple in vitro, but the current study extends this work, showing in vivo that it has anti obesity and insulin sensitizing properties in vivo. They are appropriately measured with their conclusions. I congratulate them on this nice study.

RESPONSE: We thank the reviewer for their time and appreciation of our work.

References

1. Hoch, F.L. and F.P. Hogan, *Hyperthermia, muscle rigidity, and uncoupling in skeletal muscle mitochondria in rats treated with halothane and 2,4-dinitrophenol*. *Anesthesiology*, 1973. **38**(3): p. 237-43.
2. Machado, N.L.S., et al., *A Glutamatergic Hypothalamomedullary Circuit Mediates Thermogenesis, but Not Heat Conservation, during Stress-Induced Hyperthermia*. *Curr Biol*, 2018. **28**(14): p. 2291-2301 e5.
3. Rajagopal, M.C., et al., *Transient heat release during induced mitochondrial proton uncoupling*. *Commun Biol*, 2019. **2**: p. 279.
4. Pedersen, S.B., et al., *Regulation of UCP1, UCP2, and UCP3 mRNA expression in brown adipose tissue, white adipose tissue, and skeletal muscle in rats by estrogen*. *Biochem Biophys Res Commun*, 2001. **288**(1): p. 191-7.
5. Nagase, I., et al., *Expression of uncoupling protein in skeletal muscle and white fat of obese mice treated with thermogenic beta 3-adrenergic agonist*. *J Clin Invest*, 1996. **97**(12): p. 2898-904.
6. Goldgof, M., et al., *The chemical uncoupler 2,4-dinitrophenol (DNP) protects against diet-induced obesity and improves energy homeostasis in mice at thermoneutrality*. *J Biol Chem*, 2014. **289**(28): p. 19341-50.
7. Galvao, J., et al., *Unexpected low-dose toxicity of the universal solvent DMSO*. *FASEB J*, 2014. **28**(3): p. 1317-30.
8. Chintinne, M., et al., *Beta cell count instead of beta cell mass to assess and localize growth in beta cell population following pancreatic duct ligation in mice*. *PLoS One*, 2012. **7**(8): p. e43959.
9. Yu, P.B., et al., *Dorsomorphin inhibits BMP signals required for embryogenesis and iron metabolism*. *Nat Chem Biol*, 2008. **4**(1): p. 33-41.
10. Vogt, J., R. Traynor, and G.P. Sapkota, *The specificities of small molecule inhibitors of the TGFss and BMP pathways*. *Cell Signal*, 2011. **23**(11): p. 1831-42.

14th May 2020

Dear Dr. Kirwan,

Thank you for the submission of your revised manuscript to EMBO Molecular Medicine. We have now received the enclosed reports from the referees that were asked to re-assess it. As you will see the reviewers are now globally supportive and I am pleased to inform you that we will be able to accept your manuscript pending the following final amendments.

***** Reviewer's comments *****

Referee #1 (Remarks for Author):

Thank you for carefully addressing my points.

I recommend that all the "response to the reviewer data" should be included in the manuscript. From the thermal imaging the surface temperature should be shown next to the tail temperature.

Where the energy goes in the end should be thoroughly discussed.

Referee #2 (Remarks for Author):

The authors have greatly improved the quality of their manuscript and they have replied to my comments. They have now included several key experimental controls and novel in vivo experiments.

Referee #1 (Remarks for Author):

Thank you for carefully addressing my points.

I recommend that all the "response to the reviewer data" should be included in the manuscript. From the thermal imaging the surface temperature should be shown next to the tail temperature.

Where the energy goes in the end should be thoroughly discussed.

We thank the reviewer for their contributions to our work. The thermal imaging data has now been added to the manuscript at Figure EV2G-H. We were unfortunately unable to collect surface temperature data by this method, as the heat production from brown adipose tissue confounds the skin temperature. We have also added discussion of heat dissipation as it pertains to BAM15 treatment (Lines 390-393).

Referee #2 (Remarks for Author):

The authors have greatly improved the quality of their manuscript and they have replied to my comments. They have now included several key experimental controls and novel in vivo experiments.

We thank the reviewer for their time and thoughtful consideration of our work.

The authors performed the requested changes.

Corresponding Author Name: John P. Kirwan
 Journal Submitted to: EMBO Molecular Medicine
 Manuscript Number: EMM-2020-12088